# Continual Learning with Global Alignment

**Xueying Bai**     **Jinghuan Shang**     **Yifan Sun**     **Niranjan Balasubramanian**
Department of Computer Science
Stony Brook University
{xubai, jishang, ysun, niranjan}@cs.stonybrook.edu

## Abstract

Continual learning aims to sequentially learn new tasks without forgetting previous tasks' knowledge (catastrophic forgetting). One factor that can cause forgetting is the interference between the gradients on losses from different tasks. When the gradients on the current task's loss are in opposing directions to those on previous tasks' losses, updating the model for the current task may cause performance degradation on previous tasks. In this paper, we first identify causes of the above interference, and hypothesize that correlations between data representations are a key factor of interference. We then propose a method for promoting appropriate correlations between arbitrary tasks' data representations (i.e., global alignment) in individual task learning. Specifically, we learn the data representation as a task-specific composition of pre-trained token representations shared across all tasks. Then the correlations between different tasks' data representations are grounded by correlations between pre-trained token representations. We explore different ways to learn such compositions. Without experience replay, our model achieves SOTA performance in continual learning tasks. It also achieves advanced class-incremental performance through task-incremental training. The code is available at: https://github.com/StonyBrookNLP/global-alignment.

## 1   Introduction

Continual Learning (CL) aims to develop models that can sequentially learn from streams of data and tasks, which is an important need for many real-world applications [35, 18]. One main challenge in developing CL models lies in reducing catastrophic forgetting, where models forget knowledge obtained from previous tasks after learning new tasks [38, 44, 15].

Catastrophic forgetting can happen when there is interference during task learning, especially in models that use shared parameters for all tasks [46]. Specifically, when learning a new task, if the model's gradients on the new task's loss are contradictory (e.g. in opposing directions) to those on the previous tasks' loss, the model will be updated towards a direction that increases the losses on previous tasks, causing forgetting. In this paper, we first identify factors that may lead to interference by analyzing the dot product between models' (flattened) gradients on losses from two tasks. Then, we design methods to address interference based on each factor.

Our analysis in Section 3 shows that interference mainly depends on two factors: correlations between hidden representations of the data from different tasks; and correlations between columns of the classifier (which we call class vectors) that map data representations to corresponding classes.

To address interference caused by the first factor, we want models to learn *aligned* data representations that do not have destructive correlations (i.e., leading to interference) when switching tasks and during task learning. Specifically, this requires models to accommodate future task representations when learning the current task. This motivates learning data representations based on some (well-correlated) global representations, for which we use pre-trained token representations for language data [11]. Specifically, we compose data representations as task-specific interpolations of pre-trained token

38th Conference on Neural Information Processing Systems (NeurIPS 2024).

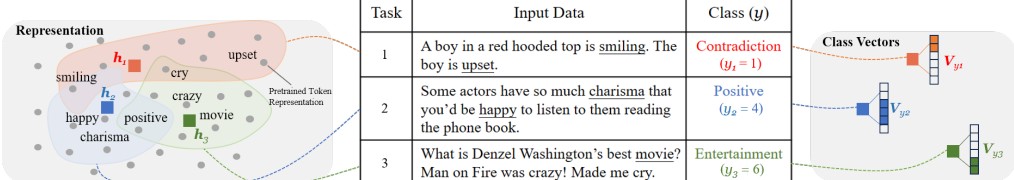

| Task | Input Data | Class ($y$) |
|------|-----------|-------------|
| 1 | A boy in a red hooded top is smiling. The boy is upset. | Contradiction ($y_1 = 1$) |
| 2 | Some actors have so much charisma that you'd be happy to listen to them reading the phone book. | Positive ($y_2 = 4$) |
| 3 | What is Denzel Washington's best movie? Man on Fire was crazy! Made me cry. | Entertainment ($y_3 = 6$) |

Figure 1: Overview of our methods. Task $i$'s data representations are denoted as $\mathbf{h}_i$ with pre-trained token representations as grey dots in the 'Representation' block. Correlations between aligned data representations from different tasks depends on correlations between pre-trained token representations. In the 'Class Vectors' block, class vectors for different classes have different focuses on representations after probing, which can reduce interference caused by overlapped representations.

representations. This allows the correlations between tasks' data representations to be grounded by correlations between pre-trained representations. We design three transformer-based [54] models that target such *global alignment*: (1). learning data representations by interpolating the pre-trained token representations through the attention mechanism; (2). the above model with additional neighborhood information to expand the search space of task information; (3). a controlled LoRA [20] model that adapts the pre-trained token representation with a small scaling factor.

To address interference caused by the second factor, we first train the classifier only when switching the task and then tune the whole model. This probing first strategy was first proposed in [27] to reduce representation distortion in single-task learning. Here we use it to reduce interference especially when there are destructive correlations between representations (e.g., caused by overlapping representations [5]). Probing enables different class vectors to focus on different features in data representations when switching tasks, which is useful when different tasks' representations are overlapped unexpectedly.

An overall view of our methods is in Fig. 1. Evaluations show that both the aligned representations and the probing first strategy improve CL performance in multiple settings. Specifically, global alignment models perform well in class-incremental evaluation after task-incremental training.

In conclusion, we make the following key contributions in this paper:
1. We identify factors that cause interference in CL, and propose to address the interference issues by learning aligned representations and applying the probing first strategy.
2. We design three models to learn aligned representations, which learn task-specific attention with different levels of adaptations on pre-trained token representations.
3. We conduct extensive experiments on multiple CL settings. Results show that our models can significantly reduce forgetting even without the use of experience replay.

## 2  Related Work

**Continual Learning**  CL Models can be divided into three main categories: regularization-based models which constrain the deviation of new parameters from the older ones [25, 62, 1, 29]; replay-based models which reduce forgetting by rehearsing on real or pseudo samples from previous tasks [35, 7] or generative models [49, 24]; and architecture-based models which learn evolving architectures for sequential tasks, with their capacities for each task carefully assigned [48, 60].

CL in NLP is an emerging area [32, 3]. MBPA++ [10] uses experience replay and local adaptation to mitigate forgetting; LAMOL [53] generates pseudo samples for replay; IDBR [21] disentangles task-agnostic and task-specific information; CTR [23] uses a capsule network for knowledge transfer. All the above models are based on pre-trained LM [11, 4, 43]. Recent works show that pre-training can alleviate catastrophic forgetting [59, 39, 28]. Our work is also based on the pre-trained LM, but we use it for alignment purposes without experience replay.

**Alignment in CL**  Recent CL works have studied the importance of alignment between different tasks' learning. Riemer et al. [46] aligns gradients between tasks to reduce destructive interference; Guo et al. [16] preserves holistic information for future tasks. If the previously learned knowledge does not align with that for future tasks, models may abruptly change previously learned knowledge and cause forgetting [5, 36, 22]. However, previous works focus on the alignment between observed tasks, which lack a global view and can cause forgetting in the future [26]. Instead, our work achieves alignment by pre-trained semantic features, which are general even to unseen future tasks.

**Adaptation Models** With limited trainable parameters, our alignment models have connections to adaptation models, which originally aimed at parameter efficiency. Different adaptation models add limited trainable parameters on the frozen transformer layer [19, 40, 17, 20]; or selectively update existing parameters [42, 61]. Recent works do adaptation by prompt tuning [31, 30, 34], which learns prompt embeddings for target tasks.

Adaptation models have also been used for CL [57, 13, 56, 45, 50]. However, most works use the models' parameter efficiency to construct progressive memory. Whether different adaptation structures influence CL, why and how they help remain unexplored. Our model has a similar form to adaptation models after derivation, but our design focuses on representation alignment rather than computational efficiency, with no progressive memory.

# 3 Problem Statement

In this paper, we focus on the catastrophic forgetting caused by cross-task interference [46, 47]. We conduct a case study to identify factors of the interference in Section 3, which motivates our global alignment approach in Section 4.

## 3.1 Continual Learning Settings

**Tasks and Data** We consider the setting where models continually learn a sequence of tasks, with the condition that the previous tasks' data becomes inaccessible when learning new tasks. We denote a representative data for each task $i$ as $(\mathbf{x}_i, y_i)$, where $\mathbf{x}_i$ is the model input and $y_i \in \mathbf{c}_i$ is its class logit. $\mathbf{c}_i$ is the set of all class logits in task $i$, and we denote the set of class logits across all $T$ tasks as $\mathcal{C} = \{\mathbf{c}_i\}_{i=1}^{T}$.

**Scenarios** We consider two CL scenarios: *task-incremental learning* and *class-incremental learning*. The main difference between them is that at inference time for task $i$ the model knows its task-specific classes $\mathbf{c}_i$ in *task-incremental learning (task-aware)*, while the model has to predict the class from all classes $\mathcal{C}$ in *class-incremental learning (task-agnostic)* [37].

**Models** Our models consist of an encoder that encodes the input $\mathbf{x}_i$ to a $d$-dimensional representation $\mathbf{h}_i \in \mathbb{R}^d$, and a matrix of class vectors (i.e., a classifier) $\mathbf{v} \in \mathbb{R}^{d \times |C|}$ whose $y_i$-th column $\mathbf{v}_{y_i}$ maps the hidden representation $\mathbf{h}_i$ to the space of the class $y_i$. The probability of $y_i$ being predicted by the model is calculated by the softmax function: $p(y_i|\mathbf{h}_i) = \text{softmax}(\mathbf{h}_i^T \mathbf{v})_{y_i}$. At training time, the softmax is computed over classes in each task, while at inference time the softmax is computed over the range of classes specified in the task- or class-incremental scenarios. We sequentially train each task with the cross-entropy loss: $\mathcal{L}(\mathbf{h}_i, y_i) = -\log p(y_i|\mathbf{h}_i)$ for task $i$'s data $(\mathbf{x}_i, y_i)$.

## 3.2 Cross-Task Interference

According to Riemer et al. [46], catastrophic forgetting can occur when a model learns a new task if its gradients on the new task's loss are contradictory (e.g. in opposing directions) to its gradients for the previous tasks' losses. In other words, the gradient descent for the new task might update the model towards a direction that increases its losses on previous tasks, and thus cause forgetting.

In this section, we analyze factors that can lead to interference between gradients. Specifically, we study a case based on a representative toy model. The model's encoder contains two linear layers with corresponding weight matrices denoted by $\mathbf{W}^l \in \mathbb{R}^{d \times d}$, where $l$ indexes the layers. The encoder outputs the representation: $\mathbf{h}_i = \mathbf{W}^2 \mathbf{W}^1 \mathbf{x}_i$ for data in task $i$. Here the input $\mathbf{x}_i \in \mathbb{R}^d$ is a $d$-dimensional vector.

Suppose we are learning task $j$ after task $i$ using gradient descent. Consider $(\mathbf{x}_i, y_i), (\mathbf{x}_j, y_j)$, two arbitrary data instances in tasks $i$ and $j$ respectively. The interference between gradients of the weight matrix $\mathbf{W}^l$ with respect to the cross-entropy loss is given by:

$$\mathcal{I}(\mathbf{W}^l) = \nabla_{\mathbf{W}^l} \mathcal{L}(\mathbf{h}_i, y_i) \cdot \nabla_{\mathbf{W}^l} \mathcal{L}(\mathbf{h}_j, y_j).$$

Destructive interference occurs when $\mathcal{I}(\mathbf{W}^l) < 0$, which can cause the model to forget task $i$'s knowledge after updating $\mathbf{W}^l$ for task $j$. On the other hand, when $\mathcal{I}(\mathbf{W}^l) > 0$, the gradients of two tasks can enhance each other which encourages knowledge transfer across tasks [46].

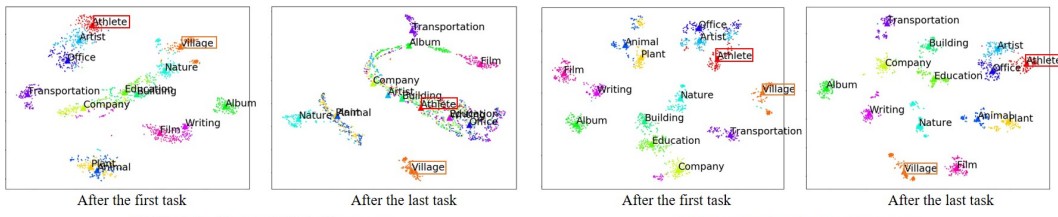

|        |        |
|:------:|:------:|
| After the first task | After the last task |
| **(a) Model without global alignment** | |

|        |        |
|:------:|:------:|
| After the first task | After the last task |
| **(b) Model with global alignment** | |

Figure 2: T-SNE plots of all tasks' data representations after learning the first (with classes *Village*, *Athlete*) and last task. Under the vanilla sequential learning in (a), after the first task, representations of data from unseen tasks are overlapped. This may cause interference when switching tasks, which makes representations indistinguishable after learning the last task. With our global alignment model (Wire-Neigh) in (b), representations remain distinguishable after the first and last tasks.

We expand $\mathcal{I}(\mathbf{W}^l)$ below. For simplicity, we calculate gradients related to the $y_i$-th and $y_j$-th class vectors in the matrix $\mathbf{v}$:

$$\mathcal{I}(\mathbf{W}^l; \mathbf{v}_{y_i}, \mathbf{v}_{y_j}) = \underbrace{\big(p(y_i|\mathbf{h}_i) - 1\big)\big(p(y_j|\mathbf{h}_j) - 1\big)}_{\text{product of probability terms} \geq 0} \underbrace{(\mathbf{h}_i^{l-1})^T \mathbf{h}_j^{l-1}}_{\substack{\text{correlation between} \\ \text{hidden representations}}} \underbrace{\mathbf{v}_{y_i}^T \mathbf{\Omega}^{l+1} \mathbf{v}_{y_j}}_{\substack{\text{correlation between} \\ \text{class vectors}}} . \qquad (1)$$

$\mathbf{h}_i^{l-1} = \begin{cases} \mathbf{W}^{l-1}\mathbf{x}_i, & l=2 \\ \mathbf{x}_i, & l=1 \end{cases}$ is the hidden representation at the $(l\text{-}1)$-th layer, $\mathbf{\Omega}^{l+1} = \begin{cases} \mathbb{I}, & l=2 \\ (\mathbf{W}^{l+1})^T\mathbf{W}^{l+1}, & l=1 \end{cases}$ is about the weight matrix at the $(l\text{+}1)$-th layer. The class vector $\mathbf{v}_{y_i} \in \mathbb{R}^d$ is the $y_i$-th column of the classifier $\mathbf{v}$, mapping the output representation to the space of class $y_i$.

Based on Eq. 1, $\mathcal{I}(\mathbf{W}^l)$ depends on the correlation between data's hidden representations at the $(l\text{-}1)$-th layer; and the correlation between class vectors transformed by weight matrices of the subsequent (e.g. $(l\text{+}1)$-th) layers. To address the interference issue, we discuss each correlation below.

**Correlations Between Hidden Representations** We consider correlations between different hidden representations at two time points (Fig. 2): (1) *The model has learned task $i$ and is switching to learn task $j$*. At this time, the model's hidden representations of task $j$'s data should not destructively interfere with those of task $i$'s data, even though the model hasn't been trained for task $j$ yet. (2) *After learning Task $j$*. At this time, the model should still produce good hidden representations for task $i$'s data, even though it no longer has access to task $i$'s training data.

Some previous works address the interference issue by forcing models to learn different tasks in orthogonal subspaces [9, 14, 55]. However, they may constrain models' knowledge transfer ability as $\mathcal{I}(\mathbf{W}^l)$ will always be zero. Other works minimize the destructive interference at time (2), by learning task $j$ with the replay of task $i$'s data [46, 47, 2]. However, if hidden representations do not correlate well when switching to task $j$, they may cause and propagate interference in task $j$'s learning (Fig. 2 (a)). Then even with replay, representations of task $i$'s data may drift towards representations of task $j$ data, leading to disruptive model updates [5].

Our work tackles correlations of hidden representations at both time points. When learning across tasks, the model is expected to produce *aligned* hidden representations of both the current and previous tasks' data at all times. The *aligned* representations should have appropriate correlations, which will not lead to destructive interference but retain the model's ability for knowledge transfer. To achieve this, we encourage the model to learn data representations for each task as different (task-specific) interpolations of pre-trained token representations. This then enables the correlations between data representations to be grounded by correlations between pre-trained token representations (details are in Section 4).

**Correlations Between Class Vectors** The interference also depends on correlations between class vectors. Assuming no shared classes in task $i$ and $j$, the class vector $\mathbf{v}_{y_i}$ is not involved in learning task $j$ and thus remains unchanged after task $i$. When learning the class vector $\mathbf{v}_{y_j}$ for task $j$, we denote $\mathbf{v}_{y_j}$ at the time step $t$ as $\mathbf{v}_{y_j,t}$. Then the correlation between class vectors $\mathbf{v}_{y_i}$ and $\mathbf{v}_{y_j,t}$ is:

$$\mathbf{v}_{y_i}^T \mathbf{v}_{y_j,t} = \mathbf{v}_{y_i}^T \mathbf{v}_{y_j,0} - \alpha \mathbf{v}_{y_i}^T \sum_t \nabla_{\mathbf{v}_{y_j}} \mathcal{L}(\mathbf{h}_{j,t}, y_j), \qquad (2)$$

where $\mathbf{h}_{j,t}$ is the output data representation at time step $t$, $\mathbf{v}_{y_j,0}$ is the initialization of the class vector $\mathbf{v}_{y_j}$, $\alpha$ is the learning rate.

In Eq. 2, the correlation of class vectors depends on the initialization of the class vector $\mathbf{v}_{y_j,0}$ and the learning of data representations $\mathbf{h}_{j,t}$ (in the gradient $\nabla_{\mathbf{v}_{y_j}} \mathcal{L}(\mathbf{h}_{j,t}, y_j)$). The learning of the representation $\mathbf{h}_{j,t}$ depends on the correlations between representations, which we have discussed above. In addition, we hypothesize that a good initialization of class vectors can help mitigate the interference problem. To obtain suitable initialization for class vectors, we apply the *probing then fine-tuning* (PF) strategy [27] which first learns the class vectors (classifier) only and then fine-tune the whole model. We describe details in Section 4.

## 4 Methodology

In this section, we introduce our models that align data representations and initialize class vectors. First, we introduce our global alignment models which learn task data representations as interpolations of pre-trained token representations. Then we discuss the probing and then fine-tuning strategy and the effects of initializing the class vectors for CL.

### 4.1 Data Representation as Interpolation of Pre-Trained Token Representations

**Pre-trained Token Representations** In this paper, we focus on language models which are typically pre-trained in a self-supervised manner [11, 41]. For example, some models are pre-trained by the masked language modeling objective, which first masks tokens in input texts and then learns models to predict masked tokens. By pre-training, models learn semantic relationships between tokens.

We consider transformer-based [54] language models. Typically, for an arbitrary task[1], the input of the model is a sequence of $n$ tokens. At the $l$-th transformer layer, denote the input representations of all tokens as $\mathbf{G}^{l-1} = [\mathbf{g}_1^{l-1}, ..., \mathbf{g}_n^{l-1}]^T$ $(l \geq 1)$ where each token representation is an $\mathbb{R}^d$ vector. Then the output token representation is:

$$\mathbf{G}^l = Attn(\mathbf{G}^{l-1}\mathbf{W}_q^l, \mathbf{G}^{l-1}\mathbf{W}_k^l)\mathbf{G}^{l-1}\mathbf{W}_v^l, \tag{3}$$

where $\mathbf{W}_q^l$ and $\mathbf{W}_k^l \in \mathbb{R}^{d \times d}$ are query and key matrices, $Attn(\mathbf{Q}, \mathbf{K}) = \text{softmax}(\mathbf{Q}\mathbf{K}^T/\sqrt{d})$ is a function calculating the attention matrix based on the query $\mathbf{Q}$ and key $\mathbf{K}$. The initial token representation $\mathbf{G}^0$ is the output of an embedding layer in the model. A feed-forward layer is applied after self-attention for token-wise transformation. We omit it here for simplicity.

After pre-training, we obtain contextual token representations $\mathbf{G}^l$ at each layer $l$ using the pre-trained matrices $\mathbf{W}_q^l$, $\mathbf{W}_k^l$ and $\mathbf{W}_v^l$. Such token representations contain semantic information of input tokens, which are general enough to accommodate diverse uses in different downstream tasks. We refer to representations $\mathbf{G}^l$ as *pretrained token representations*.

**Data Representations** To address a downstream task, models learn a data representation that summarizes the task-specific information in the entire input. Typically, these data representations are learned by fine-tuning all parameters in the pre-trained model. This type of full fine-tuning has been shown to distort the pre-trained token representations [27] and may not consider correlations between data representations across tasks. This does not fit the alignment goal stated in Section 3.2.

To align data representations across tasks, we propose to wire (interpolate) the pre-trained token representations to construct the data representation. Our proposal is based on the following hypothesis:

*Task-specific information of data can be composed from the general semantics of tokens in that data.*

For example, given the input text 'Some actors have so much charisma that you'd be happy to listen to them reading the phone book' from '*positive*' class in a sentiment analysis task, a composition of tokens {charisma, happy} can convey the information of '*positive*'.

Based on the hypothesis and the fact that pre-trained token representations contain the information of general token semantics, appropriately interpolating pre-trained token representations can represent task-specific information in the data. Since correlations between pre-trained token representations are general across tasks, this type of composition aligns data representations from different tasks.

**Alignment Effect** For an arbitrary task, the pre-trained token representations $\mathbf{G}^l$ can be interpolated to yield the data representation at the $l$-th layer as: $\mathbf{h}^l = (\mathbf{b}^l\mathbf{G}^l)^T$ where $\mathbf{b}^l \in \mathbb{R}^{1 \times n}$ is a learnable

---

[1]For an arbitrary task, we do not specify the task id $i$ by subscripts in the notations for simplicity.

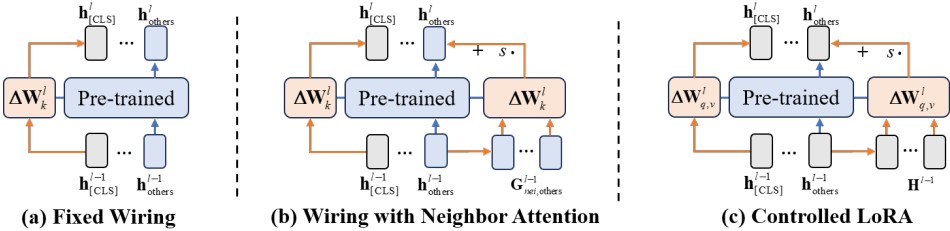

**(a) Fixed Wiring**      **(b) Wiring with Neighbor Attention**      **(c) Controlled LoRA**

Figure 3: Comparison between alignment models. Modules in blue are pre-trained and in orange are learnable. Representations in grey are mainly adapted and in blue are close to the pre-trained ones. We specify hidden representations for [CLS] and any other token as $\mathbf{h}^l_{[\text{CLS}]}$ and $\mathbf{h}^l_{\text{others}}$.

stochastic row vector with weights for the interpolation. To assess the alignment effect of this scheme, we obtain the correlation between data representations $\mathbf{h}^l_i$ and $\mathbf{h}^l_j$ from task $i$ and $j$ as:

$$(\mathbf{h}^l_i)^T \mathbf{h}^l_j = \mathbf{b}^l_i \mathbf{G}^l_i (\mathbf{G}^l_j)^T (\mathbf{b}^l_j)^T, \tag{4}$$

where $\mathbf{b}^l_i$, $\mathbf{b}^l_j$ are learned interpolations, $\mathbf{G}^l_i$, $\mathbf{G}^l_j$ are pre-trained token representations for data in task $i$ and $j$.

At any time step, the correlation between data representations is grounded by the correlation between pre-trained token representations: Eq. 4 involves the correlation $\mathbf{G}^l_i (\mathbf{G}^l_j)^T$ between pre-trained token representations; and the task-specific interpolation weights $\mathbf{b}^l_i$, $\mathbf{b}^l_j$ are also learned with the guidance of $\mathbf{G}^l_i$ and $\mathbf{G}^l_j$ respectively. This grounding to pre-trained token representations thus aligns the data representations across tasks.

## 4.2 Global Alignment Models

We develop global alignment models to learn data representations as interpolations of pre-trained token representations. Parameters in models are trained by the cross-entropy loss for each task. Following common practice in traditional models, we append a [CLS] token to the input text and use the representation of [CLS] as the data representation, denoted as $\mathbf{h}^l_{[\text{CLS}]}$.

**Fixed Wiring** Applying the interpolation weights $\mathbf{b}^l$ on pre-trained token representation $\mathbf{G}^l$ generated by Eq. 3, we have $\mathbf{b}^l \mathbf{G}^l = \mathbf{b}^l Attn(\mathbf{G}^{l-1}\mathbf{W}^l_q, \mathbf{G}^{l-1}\mathbf{W}^l_k)\mathbf{G}^{l-1}\mathbf{W}^l_v$. Since the product of a stochastic row vector and a row-stochastic matrix is stochastic, $\mathbf{b}^l Attn(\mathbf{G}^{l-1}\mathbf{W}^l_q, \mathbf{G}^{l-1}\mathbf{W}^l_k)$ can be viewed as task-specific attention on the pre-trained token representations $\mathbf{G}^{l-1}$.

Based on this, we develop a *fixed wiring* model that learns task-specific attention for [CLS] only, while using the pre-trained parameters to compute hidden representations of other tokens as the pre-trained representations $\mathbf{G}$. In the model, the task-specific attention is calculated as the attention from [CLS]'s query, using a new learnable key matrix $\Delta \mathbf{W}^l_k$ and the pre-trained query matrix $\mathbf{W}^l_q$. Formally, the data representation $\mathbf{h}^l_{[\text{CLS}]}$ is:

$$(\mathbf{h}^l_{[\text{CLS}]})^T = Attn((\mathbf{h}^{l-1}_{[\text{CLS}]})^T \mathbf{W}^l_q, \mathbf{G}^{l-1}\Delta \mathbf{W}^l_k)\mathbf{G}^{l-1}\mathbf{W}^l_v, \tag{5}$$

where $\mathbf{h}^0_{[\text{CLS}]}$ is the pre-trained embedding of [CLS]. $\Delta \mathbf{W}^l_k$ is low-ranked [20] for efficiency.

By constraining non-cls tokens' hidden representations to be close to the pre-trained token representations, the fixed-wiring model may have limited learning capacity. To avoid this, we design two other methods with improved model capacity, which we describe below.

**Wiring with Neighbor Attention** Sometimes, the task information may not be easy to extract from pre-trained representations of input tokens. For example, in a text entailment task, give a sentence pair 'The boy is crying; He's happy about the view.' with the label '*contradiction*'. The pre-trained representations of task-related tokens 'crying' and 'happy' may not be negatively correlated, which makes the model hard to learn their contradiction. However, 'crying' usually has a neighbor token 'sad', and pre-trained representations of 'sad' and 'happy' are more likely to have negative correlations. Therefore, using the information of 'sad' may make the model easier to learn the task.

Therefore, to increase the model capacity while preserving its alignment ability, we retain the guidance of pre-trained token representations while exploring the tokens' neighborhood to better search for the

task-specific information. The data representation can be written as:

$$(\mathbf{h}_{\texttt{[CLS]}}^l)^T = Attn((\mathbf{h}_{\texttt{[CLS]}}^{l-1})^T\mathbf{W}_q^l, \mathbf{G}_{expand}^{l-1}\Delta\mathbf{W}_k^l)\mathbf{G}_{expand}^{l-1}\mathbf{W}_v^l, \tag{6}$$

where $\mathbf{G}_{expand} = [\mathbf{G}^{l-1}; \mathbf{G}_{nei}^{l-1}]$ concatenates input tokens' pre-trained representations $\mathbf{G}^{l-1}$ and their neighbors' representations $\mathbf{G}_{nei}^{l-1}$.

Since each token has its own neighbors, to obtain the data representation $\mathbf{h}_{\texttt{[CLS]}}^l$ in Eq. 6, we first adapt each pre-trained token representation individually to incorporate task-specific information from their neighbors. Then we calculate task-specific attention on adapted token representations, using the attention mechanism in Eq. 5. Specifically, we adapt the $p$-th pre-trained token representation $\mathbf{g}_p^l$ by:

$$(\mathbf{g}_p^l)^T \leftarrow (1-s)\cdot(\mathbf{g}_p^l)^T + s\cdot Attn((\mathbf{g}_p^{l-1})^T\mathbf{W}_q^l, \mathbf{G}_{nei,p}^{l-1}\Delta\mathbf{W}_k^l)\mathbf{G}_{nei,p}^{l-1}\mathbf{W}_v^l,$$

where $\mathbf{G}_{nei,p}^{l-1} \in \mathbb{R}^{k\times d}$ contains $k$ neighbor representations for the $p$-th token, and $\mathbf{G}_{nei,p}^0$ is the pre-trained embedding of the neighbor tokens. $s \in \mathbb{R}$ is a scaling factor. The neighbor tokens are selected by comparing cosine similarities between token embeddings. To stay close to the pre-trained token representations but incorporate task-specific information, we update the neighbor representations as: $\mathbf{G}_{nei,p}^l = (1-s)\cdot\text{expand}(\mathbf{g}_p^l) + s\cdot Attn(\mathbf{G}_{nei,p}^{l-1}\mathbf{W}_q^l, \mathbf{G}_{nei,p}^{l-1}\Delta\mathbf{W}_k^l)\mathbf{G}_{nei,p}^{l-1}\mathbf{W}_v^l$, where $\text{expand}(\mathbf{g}_p^l) \in \mathbb{R}^{k\times d}$ is the matrix which duplicates $\mathbf{g}_p^l$ for $k$ tokens.

$\mathbf{G}_{nei}$ provides extra capacity in learning data representations. Meanwhile, the alignment effect is preserved by controlling the scale $s$, and making the neighborhood $\mathbf{G}_{nei}$ not deviate far away from pre-trained token representations $\mathbf{G}$.

**Controlled-LoRA** Another way to increase the model capacity is to adapt representations of all tokens (including both $\texttt{[CLS]}$ and other tokens in the text) by learning low-rank matrices added to the pre-trained query and value matrices (LoRA [20]). This has the model capacity close to fine-tuning, while keeping reference to the pre-trained parameters.

Denote the input representations of all tokens at layer $l$ as $\mathbf{H}^{l-1}$, where $\mathbf{H}^0 = \mathbf{G}^0$ is the pre-trained token embeddings. In LoRA, all token representations are updated by the same attention mechanism, with a learnable query and value matrices $\Delta\mathbf{W}_q^l$ and $\Delta\mathbf{W}_v^l$. The data representation $\mathbf{h}_{\texttt{[CLS]}}^l$ is:

$$(\mathbf{h}_{\texttt{[CLS]}}^l)^T = Attn((\mathbf{h}_{\texttt{[CLS]}}^{l-1})^T(\mathbf{W}_q^l + s\cdot\Delta\mathbf{W}_q^l), \mathbf{H}^{l-1}\mathbf{W}_k^l)\mathbf{H}^{l-1}(\mathbf{W}_v^l + s\cdot\Delta\mathbf{W}_v^l). \tag{7}$$

When $s = 0$, we have $\mathbf{H}^l = \mathbf{G}^l$. When $s > 0$, the added query and value matrices not only learn the task-specific attention, but also adapt the token representations which can deviate away from the pre-trained ones. To keep alignment with pre-trained token representations, we control the scaling factor $s$ to make adapted token representations close to the pre-trained ones.

### 4.3 Initialization of Class Vectors

Although data representations are grounded by pre-trained token representations, at the start time of learning task $j$ after task $i$, the interpolation for task $j$'s data may not be well learned. In this case, properly initializing class vectors of classes in task $j$ can help reduce interference.

To initialize the new class vectors when switching tasks, we adopt the *probing and then fine-tuning* (PF) strategy first proposed in Kumar et al. [27]: when learning a new task, it first freezes the encoder and only trains the classifier for the task (probing); and then tunes the encoder and classifier together (fine-tuning). This is beneficial in the case, for example, when the two tasks have similar input distributions but target different classes (e.g., news sentiment analysis vs. news categorization). In this case, data representations for two tasks may overlap when switching tasks. However, the class vectors can focus on different features in data representations after probing. Therefore, the correlation between class vectors may be small (or 0) and can reduce the interference.

## 5 Experiments

### 5.1 Datasets and Metrics

We evaluate four sequences of CL tasks: (1) **Yahoo**: a split of Yahoo dataset for news question-answer categorization [63] with 5 disjoint tasks containing 2 classes each; (2) **DB**: a split of DBPedia data for

Table 1: Results for task-incremental learning using BERT-base encoder. We report the averaged accuracy (*ACC*) and forgetting (*FGT*) with their standard deviations (*std*) on five random seeds. **Bold** scores are the best scores and underline scores are the second best. 'OOT' means out of time.

| | Model | Yahoo | | DB | | News Series | | All | |
|---|---|---|---|---|---|---|---|---|---|
| | | $ACC_{std}$ | $FGT_{std}$ | $ACC_{std}$ | $FGT_{std}$ | $ACC_{std}$ | $FGT_{std}$ | $ACC_{std}$ | $FGT_{std}$ |
| **Classifier-only** | Probing | $88.43_{0.06}$ | — | $99.30_{0.03}$ | — | $74.81_{0.46}$ | — | $89.84_{0.16}$ | — |
| **Adaptation** | FT | $73.07_{5.32}$ | $18.67_{5.41}$ | $73.15_{5.36}$ | $24.90_{5.17}$ | $59.98_{8.94}$ | $21.13_{7.44}$ | $60.92_{5.09}$ | $30.53_{4.95}$ |
| **Models** | Adapter | $79.85_{1.83}$ | $11.86_{1.83}$ | $98.70_{1.10}$ | $1.19_{1.10}$ | $65.43_{4.73}$ | $15.53_{4.29}$ | $76.31_{8.31}$ | $15.97_{8.31}$ |
| | LoRA | $86.32_{1.35}$ | $5.61_{1.35}$ | $88.63_{10.25}$ | $11.25_{10.27}$ | $69.59_{4.16}$ | $12.43_{4.14}$ | $77.37_{10.33}$ | $14.89_{10.61}$ |
| | Prefix | $89.75_{0.80}$ | $3.04_{0.79}$ | $99.83_{0.04}$ | $0.07_{0.04}$ | $75.03_{0.97}$ | $6.13_{0.98}$ | $87.53_{0.94}$ | $3.80_{0.85}$ |
| **CL** | ER | $87.42_{0.52}$ | $5.61_{0.68}$ | $91.05_{10.20}$ | $8.70_{10.14}$ | $75.47_{3.93}$ | $7.81_{5.27}$ | $66.42_{5.17}$ | $24.91_{4.60}$ |
| **Models** | A-GEM | $89.43_{0.58}$ | $2.95_{0.64}$ | $94.71_{4.70}$ | $5.98_{5.49}$ | $75.90_{3.34}$ | $6.60_{3.84}$ | $71.60_{9.38}$ | $19.40_{8.15}$ |
| | MBPA++ | $86.50_{2.23}$ | $5.30_{2.26}$ | $97.17_{3.22}$ | $2.65_{3.16}$ | $72.55_{4.12}$ | $7.23_{2.99}$ | $82.60_{0.97}$ | $9.09_{1.31}$ |
| | IDBR (-R) | $89.32_{1.17}$ | $2.19_{1.08}$ | $96.47_{4.01}$ | $3.39_{3.99}$ | $72.36_{2.20}$ | $6.50_{3.17}$ | $84.09_{0.64}$ | $7.00_{0.58}$ |
| | IDBR | $90.48_{0.55}$ | $1.32_{0.64}$ | $99.84_{0.03}$ | $0.04_{0.03}$ | $76.90_{1.98}$ | $3.24_{2.50}$ | $88.89_{0.49}$ | $2.21_{0.48}$ |
| | CTR | $87.06_{0.98}$ | $1.02_{0.74}$ | $99.04_{0.81}$ | $0.25_{0.30}$ | $75.12_{2.32}$ | $2.55_{2.19}$ | OOT | — |
| | L2P | $90.82_{0.58}$ | $0.60_{0.56}$ | $99.63_{0.36}$ | $0.29_{0.36}$ | $73.99_{2.36}$ | $3.43_{2.42}$ | $87.30_{1.60}$ | $3.30_{3.19}$ |
| | CODA | $88.33_{1.12}$ | $0.40_{0.34}$ | $99.33_{0.23}$ | $0.37_{0.25}$ | $75.13_{1.02}$ | $0.69_{0.74}$ | $87.95_{0.33}$ | $2.76_{0.38}$ |
| **Alignment** | Wire-Fixed | $91.11_{0.24}$ | $0.70_{0.28}$ | $99.85_{0.02}$ | $0.03_{0.02}$ | $76.28_{1.18}$ | $3.66_{0.84}$ | $88.63_{1.05}$ | $3.55_{1.02}$ |
| **Models** | +PF | $90.99_{0.25}$ | $0.73_{0.29}$ | $99.88_{0.01}$ | $0.02_{0.00}$ | $77.75_{1.19}$ | $1.81_{1.04}$ | **$90.18_{0.31}$** | $1.79_{0.17}$ |
| **(ours)** | Wire-Neigh | $90.98_{0.28}$ | $0.89_{0.36}$ | $99.86_{0.01}$ | $0.02_{0.01}$ | $77.10_{0.99}$ | $2.81_{0.52}$ | $88.71_{0.51}$ | $3.44_{0.51}$ |
| | +PF | $91.10_{0.19}$ | $0.63_{0.23}$ | $99.88_{0.01}$ | $0.01_{0.01}$ | $77.90_{0.45}$ | $1.98_{1.08}$ | $89.87_{0.37}$ | $2.03_{0.33}$ |
| | C-LoRA | $90.72_{0.89}$ | $1.47_{0.97}$ | $99.79_{0.05}$ | $0.11_{0.05}$ | $74.83_{2.58}$ | $5.80_{2.69}$ | $87.95_{0.97}$ | $4.71_{1.14}$ |
| | +PF | **$91.38_{0.19}$** | $0.61_{0.22}$ | **$99.89_{0.01}$** | $0.01_{0.01}$ | **$78.59_{0.85}$** | $2.42_{0.94}$ | $89.48_{0.64}$ | $3.00_{0.83}$ |
| **Non-CL** | MTL | $91.69_{0.26}$ | — | $99.61_{0.41}$ | — | $79.67_{1.99}$ | — | $90.75_{0.46}$ | — |

Wikipedia article classification [63] with 7 disjoint tasks containing 2 classes each; (3) **News Series**: a sequence of tasks on news-related data, including AG_news (news classification, 4 classes), MRPC (paraphrase detection, 2 classes) [12], RTE (text entailment, 2 classes) [58] and SST (sentiment analysis, 2 classes) [51]; (4). **All**: All tasks in the above sequences. For each task, we randomly sample 1245 samples per class, which is the least number of class samples in our datasets.

We train alignment and adaptation models in task-incremental (Task-IL) settings, where in-task classes are specified during training. Then we evaluate models on both Task-IL and Class-IL inferences, where in-task classes are not specified for Class-IL inference [37]. We measure models' average accuracy and forgetting (Appendix B) over five random seeds.

## 5.2 Models

We compare different models on the pre-trained BERT-base model, including:

**Alignment Models (ours):** (1) *Wire-Fixed*: the model freezes pre-trained token representations and learns task-specific attention for data representations (i.e. `[CLS] token`). (2) *Wire-Neigh*: the wiring model with neighbor attention. We set $s = 0.1$. For computation efficiency, we fix the number of neighbors as $k = 5$, and randomly select neighbors from top-$K$ ($K = 20$) nearest neighbors to control the range of neighborhood. (3) *C-LoRA*: the controlled LoRA model with the scaling factor $s = 0.1$. For both Wiring and C-LoRA models, we set the matrix rank $r = 8$. We also evaluate above models with the *probing then fine-tuning* (PF) strategy, denoted as *Model+PF*.

**Adaptation Models:** (1) *Fine-tuning (FT)*: fine-tuning all parameters sequentially. (2) *Prefix Tuning (Prefix)* [33]: freezing the pre-trained parameters and adding learnable embeddings to attention layers. (3) *Adapter* [19]: freezing the pre-trained parameters and injecting learnable linear projections after self-attention. (4) *LoRA* [20]: the LoRA model with suggested scaling $s = 1$ for single task learning.

**CL Models:** (1) *ER*: the FT model storing all seen examples and performs sparse (1%) experience replay. (2) *A-GEM* [7]: the FT model constraining gradients to prevent degrading performance on previous tasks. (3) *MBPA++* [10]: the FT model that stores and retrieves samples to locally adapt the model at inference time like [52]. (4) *IDBR* [21]: the FT model with information-disentanglement-based regularization and replay. We also compare to IDBR without replay, denoted as *IDBR(-R)*. (5) *CTR* [23]: an adapter-based model with capsules and task transfer routing. (6) *L2P* [57]: a prompt-based model that dynamically prompts for different data and tasks. (7) *CODA* [50]: a prompt-based model that learns attention over extensive prompt components for tasks. (8) *ERACE* [5]: a model for class-IL that calculates the current task's loss over in-task classes, while calculating the replay loss over all seen classes in the replay buffer. We also show the performance of *MTL*, which is an FT model jointly trained on all tasks (not CL). Detailed settings for models are shown in Appendix A.

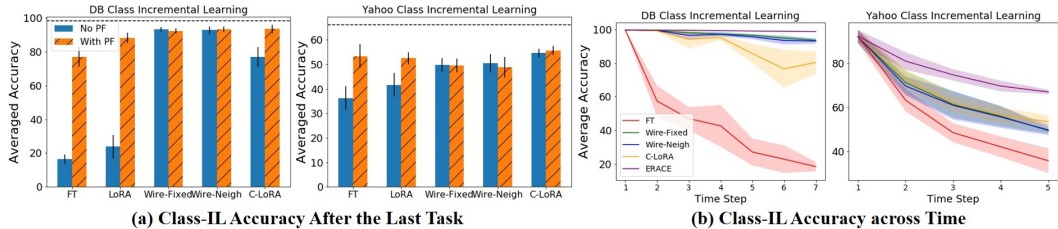

**(a) Class-IL Accuracy After the Last Task**  **(b) Class-IL Accuracy across Time**

Figure 4: (a). Class-IL accuracy after the last task. Dashed lines show accuracies of ERACE, which replays previous tasks' data with Class-IL loss. (b). Average Class-IL accuracies after each task.

Table 2: Tokens decoded from data representations by the pre-trained MLM decoder. Red tokens are tokens related to target classes in inputs; blue tokens are those related to red tokens in decodings.

| **Data** (Task 1) | *Task* | **Yahoo (news categorization, target class: Science)** | **DB (article classification, target class: Film)** |
|---|---|---|---|
| | *Sentence* 1 | what is the meaning of blog? | No One Man |
| | *Sentence* 2 | is a little space of paper of webpage space where u can write a little sentence expressing a general idea that represents what u feel or think. | No One Man is a 1932 American drama film starring Carole Lombard and Ricardo Cortez and directed by Lloyd Corrigan. It is based on a novel by Rupert Hughes. |
| **Top-10 Decoded Tokens of** [CLS] | FT | 1 A One a h An an 2 x and | . the ; : all its , his their as |
| | LoRA | Every ##nt End then X : end E G . | from to pro Sc card into B in stick long |
| | Wire-Fixed | blog ##ly user ##log Facebook blogs ##ing : Twitter web | and released , . - ( directed made / Western |
| | Wire-Neigh | blog post site posting @ web page content crawl a | . and directed is , 2 film - produced released |
| | C-LoRA | . : content blog Wikipedia posting ; # post page | is , film . of and are but - follows |

## 5.3 Results

**Alignment models are effective in Task-IL and Class-IL without replay.** Results for Task-IL are shown in Table 1. Our alignment models forget less than CL models that use experience replay during fine-tuning. They also outperform adaptation models including Adapter, uncontrolled LoRA and prefix tuning. This suggests the reduced forgetting in our alignment models is not just due to tuning fewer parameters, a feature in common with the adaptation models. In particular, C-LoRA performs better than the original LoRA model, which does not control the scaling factor for alignment. Results for Class-IL are shown in Fig. 4. On DB, alignment models achieve advanced Class-IL performance close to ERACE, a strong Class-IL model with replay, even under the Task-IL training where models are not trained on out-task classes. On Yahoo, alignment models also forget less than FT and LoRA.

**Wiring models forget less.** Without the probing and fine-tuning (PF) strategy, wiring models achieve best performances in all Task-IL experiments and Class-IL on DB (Fig. 4). We hypothesize that the good performance of wiring models is due to the strong alignment effect achieved by referencing the pre-trained token representations. With improved capacity brought by neighbors, Wire-Neigh outperforms Wire-Fixed on hard tasks like those in News Series. However, wiring models may have less capacity compared to models with more adaptations, which can limit their classification (separation) capability in Class-IL (e.g. Class-IL on Yahoo).

**PF reduces forgetting.** Fig. 4 shows that applying the PF strategy improves CL performance for all models. Although C-LoRA itself performs worse than wiring models in Task-IL, with PF it outperforms wiring models on 3/4 sequences. It also achieves best performance on Yahoo Class-IL. We hypothesize that is because PF can reduce interference when data representations have destructive correlations, as discussed in Section 3.2. Generally, PF and alignment on representations compensate each other on reducing interference. When the alignment effect is strong (e.g., wiring models), the improvement caused by PF is not very significant. However, when the alignment effect is not as strong (e.g., C-LoRA, LoRA, FT), PF can significantly improve models' performance in CL.

## 5.4 Additional Analysis and Ablations

**Interpretability of Alignment Models** Since alignment models learn data representations as the interpolation of pre-trained token representations, we interpret learned data representations using the pre-trained MLM decoder. Specifically, we decode data representations to the token space by the pre-trained decoder and then get the tokens with top-10 probabilities. This helps us understand how

Table 3: Alignment quantification on SNLI and News Series. We report Recall@20 on three random seeds.

| Model | SNLI | News Series |
|---|---|---|
| FT | $6.80 \pm 1.72$ | $7.74 \pm 3.3$ |
| C-LoRA | $14.53 \pm 0.63$ | $23.13 \pm 2.75$ |
| Wire-Fixed | $\mathbf{37.01} \pm 1.54$ | $27.80 \pm 1.48$ |
| Wire-Neigh | $36.24 \pm 1.83$ | $\mathbf{32.32} \pm 2.58$ |

data representations are closed to pre-trained token representations the model aligns to. Results are shown in Table 2. From the table, decoded tokens of alignment models are close to tokens that are related to target classes, while those of non-alignment models (FT, LoRA) are hard to interpret.

We also quantify the model's alignment ability using E-SNLI data [6], where each data's task-related tokens are annotated by human. We calculate the Recall@20 of annotated task-related tokens being retrieved from data representations, on SNLI and News Series data. The results are shown in Table 3. Results suggest that wiring models have more alignment ability than C-LoRA, in both in-task (SNLI) and CL evaluations on similar NLI tasks (News Series).

This interpretability may explain the effectiveness of alignment models in Class-IL: even when trained with local classes, data representations in each task are correlated to pre-trained token representations that relate to all tasks' classes. This may help to separate representations from different tasks.

**Influence of Scaling Factor** We show Task-IL accuracies of different scaling factors $s$ for Wire-Neigh and C-LoRA in Table 4. C-LoRA's CL performance tends to decrease when $s$ increases. This may be due to the decrease of the global alignment effect, which increases the interference. After applying PF, C-LoRA's accuracy

Table 4: Average *ACC* with different $s$ in C-LoRA, Wire-Neigh on News Series.

| Model | $s = 0$ | 0.1 | 0.4 | 0.7 | 1.0 |
|---|---|---|---|---|---|
| Wire-Neigh | 76.28 | 77.10 | 72.59 | 68.18 | 66.59 |
| C-LoRA | 74.81 | 74.83 | 72.99 | 71.02 | 69.59 |
| +PF | 74.81 | 78.59 | 77.41 | 76.83 | 76.81 |

first increases and then slightly decreases. This may be because PF reduces the interference caused by the class vectors, and the model can fully utilize its global alignment ability when increasing plasticity. However, when the scaling factor goes too large, the loss of alignment will lead to more forgetting even with PF. For Wire-Neigh, the observation is similar to C-LoRA: when $s$ goes up, the model's accuracy first increases and then decreases because of the trade-offs between global alignment and plasticity. And since Wire-Neigh interpolate pre-trained token representations with their neighbor representations, the increase of $s$ leads to the decrease of pre-trained information. And therefore we observe a more rapid performance drop when $s$ increases.

**Influence of Neighborhood** In Wire-Neigh, we randomly select five neighbor tokens from top-$K$ nearest neighbors. Here we study the effect of the range of neighborhood with different $K$ value in Table 5. For relatively simple sequences DB and Yahoo, Wire-Neigh under different $K$ has stable performance. However, for hard sequence News Series, when $K$ increases, the model has more neighbor information (more capacity) to solve the task, which first

Table 5: Average *ACC* with different neighborhood range $K$ in Wire-Neigh.

| $K$ | Yahoo | DB | News Series |
|---|---|---|---|
| 5 | 99.86 | 91.16 | 76.90 |
| 20 | 99.86 | 90.98 | 77.10 |
| 50 | 99.86 | 91.16 | 77.20 |
| 100 | 99.87 | 91.13 | 76.58 |

improves its CL performance. However, when $K$ is too large ($K = 100$), the neighbor information may become noisy, which makes the CL performance drop.

## 6 Limitation

We discuss our limitations in model and assumption perspectives: For models, without replaying previous data, there can be a problem of shifting attention; e.g. the model shifts attention on previous task-related tokens after learning new tasks. This may lead to forgetting, which we leave as our future study. For assumptions, our model assumes we have a pre-trained model. For domains that do not have well-established pre-trained semantic features, our model may not be immediately applicable. However, since fundamental models are consistently established and shown to be beneficial across different domains, we can expect our models to apply to more domains in the future.

## 7 Conclusion

In this paper, we investigate methods to address correlations between data representations and class vectors to reduce interference when training across tasks. Specifically, for alignment, we propose to learn data representations as task-specific compositions of pre-trained token representations. To learn the composed representations, we propose wiring models with or without neighbor attention and a controlled LoRA model. To address correlations between class vectors, we adopt the probing and then fine-tuning strategy, which can effectively reduce interference even when the representations do not correlate well. Experiments show that our models can successfully learn the composed representations for alignment and achieve SOTA performance in CL.

## Acknowledgments

We thank the anonymous reviewers for their insightful feedback to improve the paper. This material is based on research that is supported in part by the Air Force Research Laboratory (AFRL), DARPA, for the KAIROS program under agreement number FA8750-19-2-1003 and in part by the National Science Foundation under the award IIS #2007290.

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

# A    Detailed Experimental Settings

We provide detailed experimental settings in addition to the main paper Section 5. We train all models (except ERACE) via Task-IL training, while evaluating them in both Task-IL and Class-IL settings. We perform all experiments on one Nvidia RTX A6000 machine.

- **Probing**: We fix the encoder and only train the classifier. We train 5 epochs for each taskwith the learning rate 5e-4.
- **FT**: We fine-tune the whole model, including the encoder and classifier. We train 3 epochs for each task in BERT, with the learning rate 2e-5.
- **Adapter**: we select learning rates from {5e-5, 1e-4, 1e-3} and train {5, 20} epochs for each task. For all continual learning tasks, we train with the learning rate 5e-5 and 20 epochs. The rank of Adapter projection $r$ is 32 as suggested in the original paper.
- **LoRA (C-LoRA)**: we select learning rates from {5e-4, 1e-3} and train LoRA for {5, 8} epochs for each task.
- **Wire-Fixed and Wire-Neigh**: we select learning rates from {2e-4, 5e-4, 1e-3} and train {5, 8} epochs for each task. The rank of the learnable key matrix is 8. For Wire-Neigh, the number of neighbors is 5. In practice, they are randomly sampled from a larger neighborhood ranging from {10, 20,50,100}. We set the mixing ratio as $s = 0.1$.
- **IDBR**: We train IDBR with the learning rate 3e-5 for 3 epoches per task. We follow the k-means memory selection rule, and the replay batch size is 16 (training batch size) $\times$ number of tasks in the memory.
- **CTR**: We follow the settings in the original paper, training 5 epochs for each task.
- **L2P**: We have the prompt pool with 100 prompt tokens and select 50 of them to prepend to the input. We train the model with the learning rate 1e-3 for 20 epochs for each task.
- **CODA**: We have the prompt component size 20 for each task, and set the prompt length as 20. We train the model with the learning rate 1e-3 for 10 epochs for each task.
- **ER**: We apply sparse experience replay with 1% replay ratio. At each replay time, we sample 32 samples from the memory and perform one-step gradient descent based on them.
- **A-GEM**: We store all previous data in the memory. At each gradient step, we randomly extract 32 samples from the memory and apply the A-GEM gradient projection.
- **MBPA++**: We fine-tune the model with ER and then adapt the model at the inference time. At the inference time, we retrieve 32 nearest samples in the memory for local adaptation.

# B    Evaluation Metrics

**Recall**@$k$    Denote the set of task-related tokens of the $i$-th sample as $\text{rel}_i$, the set of top-$k$ tokens predicted from the learned data representation as $\text{pred}_i@k$. The metric Recall@$k$ calculates the proportion of task-related tokens $\text{rel}_i$ that are predicted in $\text{pred}_i@k$, which is defined as:

$$\text{Recall}@k = \mathbb{E}_i \Big[ \frac{|\text{pred}_i@k \cap \text{rel}_i|}{|\text{rel}_i|} \Big].$$

Because each data instance in E-SNLI has 5-10 task-related tokens, we use Recall@20 for evaluation.

**Average Accuracy and Forgetting**    We use the average accuracy and average forgetting similar in [8] to evaluate the performance in CL scenarios. The specific definitions are described below.

- **Average Accuracy (*ACC* ∈ [0,1])**: Let $a_{i,j}$ be the performance of the model on the test set of task $j$ after the model is trained on task $i$. The average accuracy after training on all task $T$ is:

$$ACC_T = \frac{1}{T} \sum_{j=1}^{T} a_{T,j}.$$

In this paper, we select $T$ as the end of CL task sequence.

- **Average Forgetting (*FGT* $\in$ [-1,1])**: Denote $f_{i,j}$ as the forgetting on task $j$ after the model is trained on task $i$. $f_{i,j}$ is calculated by:

$$f_{i,j} = \max_{l \in \{1,...,i-1\}} a_{l,j} - a_{i,j}.$$

And the forgetting after training on the task $T$ is:

$$FGT_T = \frac{1}{T} \sum_{j=1}^{T-1} f_{T,j}.$$

Our forgetting score divides the number $T$ of all tasks instead of $T - 1$. We do this to make the above metrics also indicate models' capacities on single tasks, i.e. single-task capacity $\approx ACC_T + FGT_T$.

## C Computation Costs

In this section, we discuss the computation costs of alignment models. All our models use shared parameters across tasks, which do not progressively increase parameters. Since the learnable new matrices in our models are all low-ranked, they require limited usage of additional memory. So we focus on the models' time consumption. For all models, we have the number of input tokens as $n$.

**C-LoRA**    Our controlled LoRA model has the same time complexity as LoRA.

**Wire-Fixed**    In the Wire-Fixed model, besides forwarding the pre-trained model to computing pre-trained token representations, we need extra computation for Eq. 5. Since we only query for the [CLS] token, the extra time complexity is $\mathcal{O}(n)$.

**Wire-Neigh**    In Wire-Neigh, the extra time consumption comes from searching for neighbors and updating the neighbor representations. We find neighbors for each token based on their embeddings at the embedding layer. The time complexity is $\mathcal{O}(Vn + VlogV)$ for computing the cosine similarity and sorting, where $V$ is the size of the token vocabulary. Then the $k$ neighbor representations are updated for each layer, with a complexity of $\mathcal{O}(nk^2)$.

In practice, since the embedding layer is fixed in our model, for each data instance we only need to find their neighbors once and then store the neighbor indices for iterative training (i.e., for several training epochs). The neighbor selection can also be accelerated by reducing the search space of neighbor tokens, for example, only searching neighbors from frequently used tokens instead of the whole vocabulary.

Wire-Neigh also needs extra memory to store neighbor representations and update them. To control the extra consumption of memory, we keep the number of neighbors as $k = 5$ and randomly sample neighbors from top-$K$ similar tokens to control the range of the neighborhood. We leave the improvement of Wire-Neigh's efficiency for future works.

**Computation induced by PF**    Another computation costs come in the probing stage, in which we fix the encoder and only train the classifier. This takes less than 40% training time (including language model forwarding) and 30% GPU memory compared to full fine-tuning.

