# OpenReview forum: "Continual Learning with Global Alignment"
_NeurIPS.cc/2024/Conference — NeurIPS 2024 poster_

### Official Review · Reviewer_R2vv · 2024-06-30

**Soundness:** 3
**Presentation:** 3
**Contribution:** 3
**Rating:** 6
**Confidence:** 5

**Summary:**

This paper tackles continual learning by addressing interference between tasks. For the interference between different tasks, the authors propose a method called ‘global alignment’ to align the data representations using task-specific compositions of pre-trained token representations. Then the authors conduct extensive experiments to verify the effectiveness of the proposed method.

**Strengths:**

1. This paper is well-written and easy to understand.

2. The motivation is clear. From the perspective of cross-task interference, this paper provides some simple but effective methods to avoid the interference of the data representation and classifier.

3. The proposed methods are easy to follow and experiments verify their effectiveness.

**Weaknesses:**

1. The analysis of interference in Section 3.2 does not consider the activation function between the two layers of the network. Multiplying the two linear weight matrices is equivalent to using only one weight matrix, so the analysis of this linear case is different from that of the non-linear case and I’m curious about the inference in the non-linear case.

2. About wiring with neighbor attention:

    a) The author needs to explain the rationality of neighborhood tokens and the impact of the size of K on the model.

    b) This method requires matching the most similar K tokens for each token given a sample. This process may require a large amount of computation.

    c) The author needs to further explain the difference between this method and controlled-LoRA. Both these two methods add a learnable term to the original token representations but adopt different generation strategies. Specifically, this method adds a term composed of neighborhood tokens, while controlled-LoRA uses low-rank matrices. So what are the advantages of this method?

3. About Controlled-LoRA:

    a) Computational burden and parameter size: As the number of tasks increases, linearly increasing model parameters is not in line with the spirit of CL, and the increased parameter size should be limited even if LoRA only adds small parameters compared to the large model.

    b) Why does Controlled-LoRA perform worse than other methods? In the case of task-IL, given the task ID for each test sample, we can directly use the best LoRA model.

**Questions:**

Please refer to Weaknesses.

---

> ### Author Rebuttal · Authors · 2024-08-07
>
> Thank you for your thoughtful review and feedback. We address your concerns as follows.
> ___
>
> **W1. The analysis of interference in Section 3.2 does not consider the activation function between the two layers of the network.**
>
> * We consider the ReLU activation which is widely used in neural networks. After ReLU, for each layer we have the output ${\bf{h}}^{(l)} = {\bf{D}}^{(l)}{\bf{W}}^{(l)}{\bf{h}}^{(l-1)}$. ${\bf{D}}^{(l)}$ is a diagonal matrix with the $i$-th diagonal value as 1 if the $i$-th element in ${\bf{W}}^{(l)}{\bf{h}}^{(l-1)}$ is positive, and otherwise 0. Then we have $\frac{\partial {\bf{h}}^{(l)}}{\partial {\bf{W}}^{(l)}} = {\bf{D}}^{(l)} \otimes ({\bf{h}}^{(l-1)})^T$ and $\frac{\partial {\bf{h}}^{(l)}}{\partial {\bf{h}}^{(l-1)}} = {\bf{D}}^{(l)}{\bf{W}}^{(l)}$ where $\otimes$ is the Kronecker product. We can calculate the gradient of each ${\bf{W}}^{(l)}$ based on the chain rule.
> * However, adding the non-linearity will complicate our analysis and also make the results hard to understand. Since it's typical to remove activation analysis for a first-pass analysis, and we do not believe the nonlinearity significantly changes the observations, we did not include it in the paper.
> ___
> **W2.a. The rationality of neighborhood tokens and the impact of neighbor size k.**
> * **Why using neighborhood tokens**: Based on Eq. 6, our model wires the pre-trained representations of input tokens to learn a task, and thus its task-learning ability depends on the range of input tokens. However, the task information may not be limited to the information of input tokens, but also their neighbor tokens. **For example**, in a text entailment task, give a sentence pair: {*s1: the boy is crying; s2: he’s happy about the view.*} with the label ‘*contradiction*’. The pre-trained representations of ‘*crying*’ and '*happy*’ may not be negatively correlated, which makes the model hard to learn their contradiction. However, ‘*crying*’ has a neighbor token '*sad*', and pre-trained representations of '*sad*' and '*happy*' are likely to have negative correlations. Therefore, introducing the information of '*sad*' may make the model easier to learn the task, and thus enhance its task learning capacity.
> * **The impact of the size of K**: Please see the general response 3.
> ___
> **W2.b. The computation cost of Wire-Neigh.**
>
> Please see the general response 2.
> ___
> **W2.c. Comparison between Wire-Neigh and Controlled-LoRA.**
>
> Please see the general response 4.
> ___
> **W3. Computational burden and parameter size of Controlled-LoRA. Why does Controlled-LoRA perform worse than other methods?**
> * a. **Controlled-LoRA does not linearly increase model parameters with tasks.** Controlled-LoRA adds low-rank query and value matrices as LoRA does. These low-rank matrices are *shared* across all tasks. We agree with your point that it is not desirable to increase model parameters when tasks grow. And that’s why we use shared parameters across tasks, and focus on alignment methods to reduce destructive interference and forgetting.
> * b. Controlled-LoRA still has forgetting because we do not progressively store parameters for each task. And it performs worse than wiring models because by learning both query and value matrices, it has weaker alignment effects. This can be compensated by applying PF (probing and then fine-tuning). After adding PF it outperforms wiring models in Table 1.

---

> > ### Comment · Reviewer_R2vv · 2024-08-13
> > **Thank you for the rebuttal**
> >
> > Thanks for answering my questions. After going through all the other reviews and the given rebuttal, I've increased my score to 6.
> >
> > W1. I understand the difficulty of the non-linearity analysis and thanks for the authors' explanation.
> >
> > W2. The comparison between Wire-Fixed, Wire-Neigh, and C-LoRA is clear, and I suggest adding this part to the main text for better understanding.
> >
> > W3. I'm sorry for the misunderstanding about C-LoRA and my concerns have been addressed.

---

> > > ### Author Response · Authors · 2024-08-13
> > > **Thank you for the response**
> > >
> > > We thank the reviewer for the thoughtful response.
> > > ___
> > >
> > > **W1 and W3.** Thank you for your understanding.
> > >
> > > **W2.** Thank you for the suggestion. We will add this part to the main paper.

---

### Official Review · Reviewer_NuNi · 2024-07-09

**Soundness:** 2
**Presentation:** 1
**Contribution:** 2
**Rating:** 5
**Confidence:** 4

**Summary:**

In Continual Learning (CL), the interference caused by the constant modification of the representation is the leading cause of catastrophic forgetting. Motivated by this and the idea that gradients in opposite directions are one cause of this interference, the authors proposed new ways of adding knowledge to a model. The authors motivate the proposal with a toy model that exemplifies where and why the interference occurs, concluding that it comes from the discrepancy between the representations of the different tasks and the relationship between the class vectors. These conclusions lead the authors to propose three proposals that slightly modify the transformer architecture, specifically adding learnable weights in the self-attention layers to generate a task-specific attention representation. The authors also propose using the probing and then fine-tuning approach presented in the past to help initialize class vectors. The results are presented in task and class incremental learning benchmarks in sequences of text datasets.

**Strengths:**

- The authors present a motivation from which they exemplify the problem in a simple way and where they want to aim their solution. This motivation may help the reader understand the context and problem to be solved.
- The paper introduces a method that alters the architecture of a transformer model. While the paper does not explicitly state this, the provided motivation helps to understand why certain parts of the self-attention layer (the k matrix) are modified.
    - The authors must describe why only the K matrix is modified and not the others.

**Weaknesses:**

- What is the actual contribution of applying wiring or C-LoRA? The results show that applying FP can benefit the methods much more than adding the proposals presented.
    - What are the implications of applying FP to the Adaptation or CL methods? The performance gain from the proposed methods is relatively low (compared to L2P), and this could further decrease with the application of FP. This requires a deeper exploration of the proposed methods and the alternatives.
- The writing is unclear and often unnecessarily complex, making reading difficult for someone unfamiliar with the subject. The notation used is only sometimes in line with the literature, which can confuse readers. In addition, there are problems explaining some terms and easily avoidable problems. For example:
    - Line 46, FP, is mentioned but needs to be correctly defined. However, the document defines the abbreviation FP multiple times after that when only one should be enough.
    - Line 182, there is an extra “192”.

**Questions:**

- My main concern is the causes of interference that motivate this work. The author hypothesised that a gradient in the same direction can alleviate forgetting between tasks, a concept also adopted by GEM, despite its susceptibility to forgetting. Even with a gradient in the same direction, if it induces significant weight changes, the representations could be compromised, leading to catastrophic forgetting.
    - How much does having a model pre-trained in tasks similar to those used in CL affect the proposed methods? The gradient will likely be low for these tasks.
    - Text benchmarks need minor modifications to the model weights due to the similar distribution between the pre-trained model and the new tasks. In cases where higher modifications are needed, can the scaling factor (s) help reduce this effect? Did you explore different values of the scaling factors?
    - Can this method work on image benchmarks? Images can have a more complex distribution than text, making avoiding interference between tasks difficult.
- Is FP applied to CL or adaptation methods in Table 1?
    - Another technique used in CL is freezing class vectors in the classifier so that only the classes present in the batch during training are modified. This is especially useful when there is no memory. Can this help further reduce interference in the proposed methods?
- Fig 2.b, what method of global alignment was used?
- Three methods to mitigate interference are presented. Can these methods be complemented by each other?
- How many neighbours are used in Wire-Neigh? Did you perform experiments to find the optimal number of neighbours?
- Did you train only the classifier for the Task Incremental Learning problem? This method should not forget, and the pre-trained model can have good prior knowledge.
- For the CIL experiments, the proposed methods have very similar results to previous methods when applying FT in Yahoo. This somewhat contradicts the claim that the proposed methods mitigate forgetting. Do you have any intuition about why FT+PF works so well?
    - In Fig 4.b., is the performance with or without FP?

**Limitations:**

As the authors mention, this method heavily relies on a pre-trained model with a similar distribution to the incoming tasks. This assumption is only sometimes true and can be more complex in scenarios with a more complex input data distribution.

---

> ### Author Rebuttal · Authors · 2024-08-07
>
> Thank you for your thoughtful review and feedback. We address your concerns as follows.
> ___
> **W1. What is the actual contribution of applying wiring or C-LoRA?**
>
> * As shown in Eq. 1, the interference depends on two factors: (1). **Correlation between representations.** The wiring and C-LoRA models are designed to address this,  by learning aligned data representations with the reference of pre-trained token representations. (2). **Correlation between class vectors.** Probing and then fine-tuning (PF) are used to address this. Both alignment models and PF are our proposals to address interference in CL.
>
> * We respectfully disagree with the point that ‘*applying PF can benefit the methods much more than adding the proposals presented*’.  Without PF, wiring models can already achieve superior performance (Table 1). PF is especially helpful when the alignment between representations is not strong, as discussed in paper l327 -l330.
>
> * About the comparison to L2P, almost all of our models, with and without PF, achieve better performance than L2P in Table 1. If we misunderstood your point, please let us know and we will be happy to discuss it.
>
> ___
> **Q1.a. Gradient direction and significant weight changes.**
>
> * The interference problem which focuses on the direction of gradients is fundamental in CL, as studied by many previous works like GEM you mentioned. The weight change is another important problem. We believe these two do not contradict each other.
>
> * The weight changes depend on both the learning rate and the gradients. In our experiments, models with similar structures (e.g. models with pre-trained encoder frozen) are tested with similar learning rates for a fair comparison.
> ___
>
> **Q1.b. Effect of having a model pre-trained in tasks similar to those used in CL.**
>
> * **Pre-trained and downstream CL tasks**: In pre-training, models are pre-trained in a self-supervised manner (masked language modeling), which does not have implications for any supervised task used in CL.
>
> * **Gradient will likely be low**: In downstream tasks for CL, the task data usually have different distributions from that in pre-training (e.g., detecting text entailment of sentence pairs), and the classifier for the task has to be learned from random initializations. Both of these will make the gradient not low (i.e., the loss is not small) at the beginning of tuning.
>
> * **Use of pre-trained model in our work**: Our work does not hypothesize the similarity between pre-training and fine-tuning tasks. On the other hand, we use pre-trained token representations as a basis/reference, and the models have to learn task-specific attention to them (with the randomly initialized key matrix). A better pre-trained model can help if it learns true correlations between pre-trained token representations, which will provide better pre-trained token representations for better alignment effects in our models.
> ___
> **Q1.c. In cases where higher modifications are needed, can the scaling factor(s) help reduce this effect?**
>
> We assume the ‘higher modifications are needed’ means we need more plasticity to learn every single task. In this case, we can increase the scaling factor in Controlled-LoRA or expand the range of neighbor tokens in Wire-Neigh.
>
> The scaling factor controls the balance between the alignment effect and the models’ tuning capacity. If the datasets are hard and we need more plasticity, then increasing the scaling factor will help. However, when the scaling values become too large, the alignment effect will be reduced and the model may have more risk of forgetting as well. For comparison, please refer to the results of C-LoRA (with the scaling factor 0.1) and LoRA (with the scaling factor 1) in Table 1. Although LoRA achieves better single-task performance than C-LoRA, it also forgets more in our CL experiments.
> ___
> **Q2. Is FP applied to CL or adaptation methods in Table 1?**
>
> * No. Since PF is proposed based on our analysis in Eq. 1, applying it to CL is also a part of our proposed models. So we did not apply it to other baselines.
>
> * **Modify only the classes in the batch**: Our standard training process has already used the mentioned strategy. In our training, for each task, only the class vectors of classes presented in that task are trained. For each batch, data are randomly sampled over all classes in the task. Besides this, our proposed model can achieve additional improvements in CL by the global alignment design and PF.
> ___
> **Q3. Fig 2.b, what method of global alignment was used?**
>
> Sorry for the confusion. Fig 2.b is plotted under the wiring model with neighbor attention.
> ___
> **Q6. Did you train only the classifier for the Task-IL?**
>
> Yes. We have provided this result in the paper, row for ‘Probing’ (above adaptation models) in Table 1. The pre-trained model has good prior knowledge in DB, but the prior knowledge is not sufficient for Yahoo and News Series as there are large performance gaps between MTL and Probing. And all of our methods outperform the probing performance in Table 1.
> ___
> **Q7. Why FT+PF works so well?**
>
> * As shown in Eq. 1, both correlations between data representations and class vectors will affect the interference. And PF can reduce the interference caused by class vectors. This may be the reason that using FT + PF works well, as discussed in l323 - l330.
> * **Why Alignment models + PF have similar results to FT + PF**: for CIL on DB, alignment models outperform FT + PF. For CIL on Yahoo, models may need more separation ability in each task's learning. Since FT has a strong separation ability for each task, it may have potential in CIL with PF. Similarly, C-LoRA has more single-task capacity than wiring models and also has alignment effects, so it achieves the best results in CIL on Yahoo.
> * **Fig 4.b.**: The performance is without PF, since wiring and C-LoRA themselves perform well in Fig 4.a.
> ___
> For questions Q4 and Q5, please refer to the general response 4, 3.

---

> > ### Comment · Reviewer_NuNi · 2024-08-09
> >
> > I thank the authors for their detailed answers.
> >
> > W1. Yes, I am sorry about my error. I associate IDBR's results with L2P, and because IDBR uses memory, I understand that their comparison is not completely fair.
> >
> > For Q1.a, I was referring to the gradient in the model. I understand that the gradient in the classifier can be high, and it should be high to learn the downstream task. However, some techniques have been proposed to mitigate the classifier's forgetting. For example, fix the columns of those classes not present in the batch (to mitigate the interference at the gradient level).
> >
> > Q1.c. Yes, this is exactly what I was referring to. If the task distribution is very different, the strategy must modify the model weights, increasing the scaling factor. I understand that this can increase forgetting, but it is interesting to understand how much the scaling factor can move to learn a very out-of-distribution downstream task but mitigate forgetting. This is the main challenge in CL when using pre-trained models.
> >
> > Q.6. Thanks. I didn't see it. Would it be a good idea to add something in the first column?
> >
> > Due to this, I will increase my score to a 5.

---

> > > ### Author Response · Authors · 2024-08-10
> > >
> > > We thank the reviewer for the thoughtful response and detailed explanation.
> > > ___
> > > **W1.** Thank you for your understanding.
> > > ___
> > > **Q1.a.** We agree that fixing the columns of classes not present in the batch can mitigate interference *in the classifier*. Under this setting, updating the columns of class vectors from new tasks will not influence the class vectors learned in the previous tasks (i.e., zero gradient for class vectors of previous tasks). As stated in the rebuttal Q2,  we have already used this technique in training all our models, including baselines.
> > >
> > > However, this technique can not fully mitigate interference *in the encoder*. Since the encoder is shared across all tasks, the value of new class vectors will influence the gradients on the encoder (Eq. 1), in both the magnitude and direction. Since the loss is usually non-zero at the beginning of task learning, the gradients of the encoder are unlikely to be zero*. This will cause interference based on the encoder’s gradients, on which we focus in this paper.
> > >
> > > *If we have a good pre-trained model that can solve the target task by only tuning the classifier, then the gradients on the encoder are zero (i.e., the loss is zero) after probing. However, that is not always the case as shown in the probing results of Table 1.
> > > ___
> > > **Q1.c.**  Thanks for your suggestions. We show the Task-IL accuracies of different scaling factors $s$ for controlled-LoRA (C-LoRA) and Wire-Neigh in the tables below.  We test on the News-Series sequence, where the tasks are from nature language inference (NLI) datasets that have different data distributions from pre-training. The probing performance on News Series also has a large gap to MTL, which indicates that models need more plasticity to solve the tasks.
> > >
> > > | Model | s = 0 (Probing) | s = 0.1  |s = 0.4|s = 0.7|s = 1.0|
> > > |-|-|-|-|-|-|
> > > |  C-LoRA  |   74.81  |   74.83  |72.99|71.02|69.59|
> > > |  C-LoRA + PF  |  74.81 |   78.59|77.41|76.83|76.81|
> > >
> > > * For C-Lora, when $s$ increases, the model's global alignment effect decreases while the plasticity increases. The results suggest: (1) C-LoRA’s CL performances decreases when $s > 0.1$. This may be because the encoder loses the global alignment effect, which also misleads the learning of the classifier and increases the interference. (2) After applying PF, C-LoRA’s accuracy first increases and then slightly decreases, overall consistently outperforms probing. This may be because PF reduces the interference caused by the class vectors, and the model can fully utilize its global alignment ability when increasing plasticity. However, when the scaling factor goes too large, the loss of alignment ability will lead to more forgetting even with PF.
> > >
> > > | Model | s = 0 (Wire-Fixed) | s = 0.1  |s = 0.3|s = 0.5|
> > > |-|-|-|-|-|
> > > |  Wire-Neigh  |   76.28|77.20 |75.19|72.46|
> > >
> > > * For Wire-Neigh, we can improve its plasticity by expanding the neighborhood (general response 3), or increasing the scaling factor $s$ that represents the interpolation between the pre-trained token representations and their neighbor representations. The observation is similar to C-LoRA: when $s$ goes up, the model’s accuracy first increases and then decreases because of the trade-offs between global alignment and plasticity.
> > > ___
> > > **Q6.** Yes, we will mark the probing as the classifier-only learning baseline in the first column.

---

### Official Review · Reviewer_Ep6T · 2024-07-09

**Soundness:** 3
**Presentation:** 4
**Contribution:** 3
**Rating:** 7
**Confidence:** 3

**Summary:**

This paper addresses the problem of Task-Incremental-Learning (TIL) with pre-trained transformer in the context of NLP. The author extended their experiments in the Class-Incremental Learning (CIL) scenario. The authors identify potential forgetting causes as (1) negative correlation between data representations and (2) negative correlation between class prototypes. After theoretically justifying their claim, the authors propose to align the model representations with either a) learning new key matrices for pre-trained queries and values b) learning new key matrices for with additional token chosen as the nearest neighbors of the considered token c) learning new queries and values from the original ones with a low-rank adaptation strategy. The author additionally align prototypes (class vectors) leveraging a probing and fine-tuning strategy. This paper shows superior performances on various dataset in CIL and briefly discuss the effect of various components.

**Strengths:**

- clear and understandable paper
- the equations are well derived and clear
- the proposed approach has interesting theoretical justifications
- The obtained performances on the TIL settings are compelling

**Weaknesses:**

1. While I believe the evaluation to be sufficient, it would improve the paper to include more recent prompt-learning techniques such as CODA [1] which showed stronger performances than L2P. I believe it would equally be interesting to see this approach applied to Computer Vision problems, even though I understand this paper focuses on NLP.

2. Although the proposed approach is theoretically justified, it would be interesting to quantify such alignment through experiments, by computing data representations covariance/correlation between transformed class vectors; for each alignment strategy.

3. I appreciate the effort to include results in CIL scenarios. However, I think more discussions as to how the proposed approach performs poorly compared to ER-ACE could be introduced.

4. What is the link between the findings of this paper and previous work on orthogonal subspaces in continual learning [2, 3]? In these works, amongst others, it seems that the correlation between hidden representation should be zero, and not positive. I think such work should be included in related work section.

5. A discussion regarding the extra computation induced by the alignment strategies and the PF would be welcomed as well, as it seems to increase it considerably.

6. The code is unfortunately not accessible to the reviewers.

If the authors can address most of the above points I would happily increase my score.

**Typos and presentation**

- LoRA l44 and PF l.46 are not defined. Please either define the acronyms or cite the corresponding paper.
- l.114 $h_{\tau}$ should be h_{i}
- I do not like the use of RHS l.163 and 164.
- l. 182: "grounded192"?

[1] Smith, James Seale, et al. "Coda-prompt: Continual decomposed attention-based prompting for rehearsal-free continual learning." Proceedings of the IEEE/CVF Conference on Computer Vision and Pattern Recognition. 2023.

[2] Wang, Xiao, et al. "Orthogonal subspace learning for language model continual learning." Findings of the Association for Computational Linguistics: EMNLP 2023, pages 10658–10671

[3] Chaudhry, Arslan, et al. "Continual learning in low-rank orthogonal subspaces." Advances in Neural Information Processing Systems 33 (2020): 9900-9911.

**Questions:**

See weaknesses.

**Limitations:**

Limitations have been partially addressed in the main draft. I believe a discussion regarding the poor performances on CIL scenarios should be included, as well as a discussion on the potential computation overhead.

---

> ### Author Rebuttal · Authors · 2024-08-07
>
> Thank you for your thoughtful review and feedback. We address your concerns as follows.
> ___
> **W1. Comparison to more recent prompt-learning techniques and the application to CV tasks.**
>
> * Comparison to CODA: we show CODA's average accuracy on Task-IL below. Since CODA's hyperparameters are set for CV tasks, the recommended settings may not be optimal for our NLP evaluation. This may be the reason that CODA does not outperform L2P in our case. However, after finding the optimal NLP hyperparameters CODA may perform better than L2P as shown on CV tasks. We leave this into our future work. And we will add more recent prompt-learning techniques to our related works.
> | Model | DB | Yahoo |News Series
> |-|-|-|-|
> CODA|98.67|88.05|75.07
> L2P|99.63|90.82|73.99
>
> * Application to CV tasks: please see the general response 5.
> ___
> **W2. Quantify alignment through experiments.**
> * **Representations' correlation between transformed class vectors**: By evaluating our Task-IL models in the Class-IL evaluation, the Class-IL results show the correlation between learned representations and class vectors. If data representations and class vectors are not correlated well across tasks, then models may fail to assign data representations to corresponding classes (across all tasks). Our models have improvements in Class-IL as well, which suggests they have learned representations correlated to all class vectors.
> * **Representations' correlation in the pre-trained space**: To further evaluate the correlation between data representations, we decode representations to the token space using the pre-trained decoder. If data representations are well correlated and guided by the pre-trained representations, they should be decoded to tokens that are related to tasks, as shown in paper Table 2. We quantify the model's alignment ability using E-SNLI [1] data, where the data's task-related tokens are highlighted by human annotators. We calculate the Recall@20 of task-related tokens decoded after training on News Series and SNLI (single task). The results are shown below.
> | Model | SNLI |News Series
> |-|-|-|
> FT|6.80|7.74
> C-LoRA|14.53|23.13
> Wire-Fixed|37.01|27.80
> Wire-Neigh|36.24|32.32
>
> Results suggest that wiring models have more alignment ability than C-LoRA, in both in-task (SNLI) and CL evaluations on similar NLI tasks (News Series). And Wire-Neigh has a better alignment ability in CL evaluations.
> ___
> **W3. How the proposed approach performs poorly compared to ER-ACE.**
> * ER-ACE replays previous tasks’ data at each training step. This means it is learned to explicitly distinguish classes from different tasks, which is effective for class-IL. However, our methods are only trained in a task-IL manner without experience replay, which means (1). They have no information about classes in previous tasks; (2). They do not have access to previous data so they are not explicitly trained to distinguish classes from different tasks.
> * We believe it is unfair to directly compare our models to ER-ACE since ER-ACE uses more information for class-IL during training. However, even under this setting, our models perform close to ER-ACE on DB, and achieve good class-IL performance on Yahoo compared to fine-tuning. Their ability to separate representations from different tasks/classes is a result of our global alignment design.
> ___
> **W4. What is the link between the findings of this paper and previous work on orthogonal subspaces in continual learning?**
> * Thanks for pointing out these related works. As you mentioned, they focus on getting zero interference between gradients, which minimizes the interference but also reduces the chances of the gradients being in the same direction and enhancing different tasks’ learning.
> * Our work does not restrict the gradients to be orthogonal. On the other hand, we guide the gradients by the pre-trained representations. If there are two tasks that have positive knowledge transfer between them, models may focus on tokens that have positively correlated pre-trained representations, which make the inner product between their gradients positive and enhance the two tasks’ learning (if the class vectors are not negatively correlated).
> ___
> **W5. Extra computation induced by the alignment strategies and the PF.**
>
> Please see the general response 2.
> ___
> **W6. The code is unfortunately not accessible to the reviewers.**
>
> We are cleaning the code and will send the link to AC soon.
>
> **Typos and presentations.**
>
> Thanks for pointing out the typos and the presentation suggestions. We will correct and modify the paper accordingly.
> ___
> [1] Camburu et al. e-SNLI: Natural Language Inference with Natural Language Explanations. NIPS 2018.

---

> > ### Comment · Reviewer_Ep6T · 2024-08-11
> > **Thank you for the rebuttal**
> >
> > I thank the authors for taking the time to answer my question and improve my understanding of the manuscript.
> >
> > **W1**
> >
> > I believe it would make it a fairer comparison to include CODA as a SoTA prompt-learning based method after hyper-parameter, since it outperforms L2P on vision tasks, at least for CIL scenarios. I acknowledge that results might differ for the TIL scenario. I still appreciate the effort and I understand that the limited time for rebuttal is not necessarily sufficient to conduct an extensive hyper-parameter search.
> >
> > **W2**
> >
> > Thank you for the extra experiments, which are convincing. Including them in the main draft could also clarify the impact of the proposed loss on the model alignment.
> >
> > **W3**
> >
> > Thank you for the clarification. I agree that direct comparison to replay-based methods might be unfair.
> >
> > **W4**
> >
> > Thank you for the clarification. I would advise the authors to include such discussion in the related work.
> >
> > **Time consumption**
> >
> > Thank you for the additional information. Such details could be included in the paper or the appendix, as computation is a major focus focus in Continual Learning.
> >
> > Overall, I do not have major concern and I will **increase my score to 7**.

---

> > > ### Author Response · Authors · 2024-08-11
> > > **Thank you for the response**
> > >
> > > We thank the reviewer for the thoughtful response and helpful suggestions.
> > > ___
> > > **W1**
> > >
> > > We agree with that. We are working on the hyperparameter searching on CODA and will include it for comparison in the later version of the paper.
> > >
> > > **W2**
> > >
> > > Thank you for the suggestions. We will include them in the main draft.
> > >
> > > **W3**
> > >
> > > Thank you for your understanding.
> > >
> > > **W4 and Time consumption**
> > >
> > > Thank you for the suggestions. We will include the discussions in the paper.

---

### Official Review · Reviewer_LhK3 · 2024-07-11

**Soundness:** 3
**Presentation:** 3
**Contribution:** 3
**Rating:** 6
**Confidence:** 3

**Summary:**

This paper studies the cause of cross-task interference in class-incremental learning of transformer-based language models. The authors disentangle the cause into the correlation (i) between data representations and (ii) between class vectors in the linear classifier. To tackle (i), the authors propose three ways to construct data representations at each layer by learning an attention over the pretrained token representations. To tackle (ii), the authors propose to only train the classifier for a new task (to obtain a good initialization) before jointly training both the classifier and the encoder.

The authors perform experiments with the pretrained BERT-base model and various text datasets. Training is in the task-incremental setup (where task labels are provided and the model predicts over in-task classes); the model is evaluated on both task-incremental and class-incremental (where task labels are not provided and the model predicts among all classes) setups. The authors found that their methods, "alignment models", outperform existing adaptation and continual models.

**Strengths:**

1. This paper is well-written and I did not find major technical flaws.

2. I enjoyed reading the motivation of the paper in Sec. 3 where the authors examine the causes of cross-task interference.

3. I find that the initialization of class vectors may influence cross-task interference interesting, although I have some related questions.

**Weaknesses:**

1. While the main goal of the paper is reducing cross-task interference, the main results (Table 1) is using the task-incremental setup, where task labels are known during prediction. The class-incremental learning results are only in Fig. 4, comparing the proposed method only with LoRA and ERACE.

2. I find it a bit hard to infer where the performance improvement comes from from the results. I wonder if it is possible to do some fine-grained ablation analysis that verifies that the proposed method indeed helps by reducing overlap in data representations and in class vectors' features for different tasks. For example, one could measure the accuracy on the task level or look at the confusion matrix, which may reveal some information about cross-task confusions.

3. Is $\Delta W$ shared across tasks? If so, I don't understand why the proposed method does not forget. The authors hypothesize that this is potentially due to referencing pretrained representations. However, since the [CLS] representations are constructed involving $\Delta W$, I'd expect the model to lose some ability to generate good representations for past tasks. Could the authors elaborate on this point?

**Questions:**

1. Could the authors explain the choice of training in TIL while evaluating in CIL, as opposed to training and evaluating in the same setup?

2. (Eq. 2) Since weights are usually initialized from a distribution centered around 0, shouldn't the first term be quite small?

3. (Sec 4.3) Does probing perform softmax over only the new classes? If so, how does it help two class vectors for different tasks to focus on different features, since the loss does not require differentiating between the two classes?

4. (Sec. 4.2) In the wiring models, what is the intuition behind only replacing the key matrix but not the query and value matrices?


Minor:

5. (Eq. 1) It seems that interference is eliminated once any of the three components is 0. Is it possible to fix the class vectors to be the canonical basis which are guaranteed to be perpendicular, and only learn the data representations?

6. (Sec. 4.2) In Wire-Neigh, do you need to find the $K$ nearest neighbors for each token? If so, how expensive it is in practice?

7. (L#219) Could you elaborate on what "grounded" means here?

8. Missing references: [1, 2] are two prompting-based CL methods that are shown to perform better than L2P.

9. (L#41-44) I recommend changing the indices to, e.g., "(i), (ii), (iii)", as a different "(2)" is referred to in the next paragraph.

10. (L#53) "by" -> "after"

11. (L#182) "grounded192" -> "grounded"


[1] DualPrompt: Complementary Prompting for Rehearsal-free Continual Learning. Wang et al. ECCV 2022.

[2] CODA-Prompt: COntinual Decomposed Attention-based Prompting for Rehearsal-Free Continual Learning. Smith et al. CVPR 2023.

**Limitations:**

The authors have addressed the limitations clearly in Sec. 6.

---

> ### Author Rebuttal · Authors · 2024-08-07
>
> Thank you for your thoughtful review and feedback. We address your concerns as follows.
> ___
> **W1. While the main goal of the paper is reducing cross-task interference, the main results are using the Task-IL setup.**
>
> We focus on Task-IL because:
>
> * Cross-task interference is a fundamental problem in Task-IL, which focuses on the gradients of shared parameters (i.e., the encoder) across all tasks. If destructive interference happens, the encoder may be updated to generate drifted representations for previous tasks, and the previous classifier may not be able to distinguish them (i.e., forget previous knowledge). Previous works study interference over the Task-IL setting as well, as cited in the paper.
>
> * Task-IL focuses on the interference in the shared parameters, while Class-IL further requires distinguishing between classes from different tasks. Although our paper shows they are inherently related, the distinction between cross-task classes is not our original goal.
>
> We do not compare with more Class-IL methods because they are explicitly designed to distinguish cross-task classes during training like ERACE, while we only use Task-IL training without information of previous tasks’ classes. However, our method can be combined with those methods to achieve stronger performance in Class-IL.
> ___
> **W2. Ablation analysis of proposed models in reducing overlap in data representations and class vectors for different tasks.**
>
> * In paper Fig. 2, we show a T-SNE plot of data representations on DB sequence for our model and the fine-tuning model. Our model generates non-overlapping representations after the first and last task. However, fine-tuning representations overlap after the first task, and further mix up after the last task.
>
> * By evaluating our Task-IL models in the Class-IL evaluation, the Class-IL results show the separation of learned representations and class vectors. If data representations and class vectors are not separable enough across tasks, then models may fail to assign data representations to corresponding classes (across all tasks). Our models have improvements in Class-IL as well, which suggests they reduce overlapping in representations and class vectors.
> ___
> **W3. Is $\Delta W$ shared across tasks?**
>
> Yes,  $\Delta W$  is shared across all tasks. The proposed model still has forgetting as shown in experiments. However, the forgetting is significantly reduced by using our global alignment methods, which results in less interference during training and thus less forgetting.
> ___
> **Q1. The choice of training in TIL while evaluating in CIL.**
>
> * For Class_IL, without replay or pre-separating class vectors, training with the loss on in-task classes can reduce forgetting compared to the loss over all classes [1]. The reason is that, if we use the loss on all seen tasks' classes but do not have data from all those classes (since no replay), it will cause data imbalance on classes during training. This may distort previously learned representations and cause forgetting.
>
> * As mentioned in **W2**, this setting can also evaluate the separation of data representations and class vectors learned in Task-IL.
> ___
> **Q2. Influence of the first term in Eq. 2 when the weights are usually initialized to be centered around 0。**
>
> We write Eq. 2 with the learning rate $\alpha$ as:
> ${\bf{v}}\_{y_i}^T{\bf{v}}\_{y_j} = {\bf{v}}\_{y_i}^T{\bf{v}}\_{{y_j},0} - \alpha{\bf{v}}\_{y_i}^T\sum\nolimits_{t}\nabla_{{\bf{v}}\_{y_j}} \mathcal{L}({\bf{h}}\_{j, t}, y_{j})$.
>
> * Whether the first term is small depends on the class vector learned from the previous task, i.e., ${\bf v}\_{y_i}$.
>
> * The influence of the first term also depends on the scale of the second term. Since models are fine-tuned with relatively small learning rates like $\alpha$ = \{1e-3, 1e-4, 1e-5\}, the second term is also 'small'. So the first term can still have a large impact even if the class vectors are initialized centered around 0.
> ___
> **Q3. How does probing help two class vectors for different tasks to focus on different features?**
>
> In the probing stage, we fix the encoder learned from previous tasks and only train the classifier for the new task. Therefore, the encoder generates the new task’s representations based on the previous task’s knowledge. And the class vectors of new and previous tasks can have different focuses on such knowledge, based on different objectives.
> ___
> **Q4. Replacing the key but not the query and value matrices.**
>
> Please see the general response 1.
> ___
> **Q5. Interference is eliminated once any of the three components is 0. Is it possible to fix the class vectors to be the canonical basis?**
>
> * Our goal is to avoid destructive interference when learning across tasks, which encourages both zero and positive interference at adequate times. If we fix the class vectors to be perpendicular to each other, there will be no positive interference that can help positive knowledge transfer.
>
> * Also, the perpendicular class vectors may not be good class vectors that prevent feature distortions when tuning a pre-trained model. This will cause the loss of pre-trained knowledge during learning and make models lose generalization on OOD data, which can also make models perform inferior in CL.
> ___
>
> **Q6. Computation cost for Wire-Neigh.**
>
> Please see the general response 2.
> ___
>
> **Q7. 'Grounded' in L219.**
>
> “Grounded” means that the correlations between data representations are decided by the correlations between corresponding pre-trained representations $G_i$, $G_j$, and the attention on these representations (learned in tasks). For example, if the two tasks pay attention to tokens that have orthogonal pre-trained token representations ($G_iG^T_j = 0$), e.g. when the two tasks’ information is irrelevant, then the alignment models will generate orthogonal data representations as well.
> ___
> [1] Masana et.al. Class-incremental learning: survey and performance evaluation on image classification. TPAMI, 2022.

---

> > ### Comment · Reviewer_LhK3 · 2024-08-10
> > **Thanks for rebuttal**
> >
> > I thank the authors for answers my questions. My concerns are resolved and I've increased my score to a 6.
> >
> > Regarding W3, I wonder if the reason that a shared $\Delta W$ shows low forgetting could also be due to that it does not contain a lot of parameters. This recent paper [1] shows that PEFT methods enjoy low forgetting when you don't have too many tunable parameters.
> >
> > [1] Thede et al. Reflecting on the State of Rehearsal-free Continual Learning with Pretrained Models. CoLLAs 2024.

---

> > > ### Author Response · Authors · 2024-08-10
> > >
> > > We thank the reviewer for the thoughtful response and helpful reference.
> > > ___
> > > Yes, we agree that tuning fewer parameters may help to mitigate forgetting. In the paper Table 1, we observe popular PEFT methods (Adapter, LoRA, Prefix-Tuning) have overall less forgetting than pure fine-tuning (FT).
> > >
> > > In this paper, we provide an angle of interference that connects PEFT models’ superior performance to a global alignment effect. According to [1], token representations in different PEFT models can be viewed as combinations of the pre-trained token representations and some modification vectors. When the models have limited parameters, the modifications may also be limited (e.g., prefix tuning with a limited number of prompts). This makes the token representations strongly guided by the pre-trained token representations, which has the global alignment effect to reduce interference and mitigate forgetting.
> > >
> > > However, if models have too few parameters, they may lose the plasticity to learn hard tasks and perform inferiorly in CL. Therefore, we also study ways to reduce forgetting when increasing model parameters/adaptations for plasticity.
> > > ___
> > > [1] He et al, Towards a Unified View of Parameter-Efficient Transfer Learning, ICLR 22

---

### Author Rebuttal · Authors · 2024-08-07

We thank all reviewers for their thoughtful reviews and feedback. We address common questions as follows.
___
**1. The intuition behind only replacing the key matrix but not the query and value matrices in wiring models.**
* **Why not replace value matrices**: when replacing the value matrices with $\Delta {\bf W}_v$, the data representation ${\bf{h}}^{(l)}$ turns out to be:  $({\bf{h}}^{(l)})^T = ({\bf{b}}^{(l)})^T{\bf{A}}^{(l)}{\bf{G}}^{(l-1)}({\bf W}\_v^{(l)} + \Delta {\bf W}^{(l)}_v)$. Compared to our goal in Eq. 6, this will reduce the guidance of pre-trained token representations, which does not fit our wiring goal.
* **Why not replace query matrices**: since the wiring models only relearn the attention for [CLS] token, for all tasks they only query from [CLS] (other tokens' queries are pre-trained). However, the keys are from all tokens in the input and are different when tokens from different tasks have different distributions. Intuitively, we learn the key matrices applied over all input tokens instead of the query matrices applied on the same [CLS] across different tasks.
___

**2. Extra computation induced by the alignment strategies and the PF.**
* **Computation induced by searching for neighbors in Wire-Neigh**: We find $k$ neighbors for each token based on their embeddings at the embedding layer. The time complexity is $O(vn)$ where $v$ is the size of vocabulary and $n$ is the size of input tokens. Then the neighbor representations are updated for each layer, with a complexity of $O(nk^2)$.
In practice, since the embedding layer is fixed in our model, for each data instance we only need to find their neighbors once and then store the neighbor indices for iterative training (i.e., for several training epochs).
The neighbor selection can also be accelerated by reducing the search space of neighbor tokens, for example, only searching neighbors from frequently used tokens instead of the whole vocabulary.

* **Computation induced by PF**: in the probing stage, we fix the encoder and only train the classifier. This takes less than 40% training time (including LM forward) and 30% GPU memory compared to full fine-tuning.
___

**3. Impact of hyperparameter K in Wire-Neigh.**

For computation efficiency, we fix the number of neighbors as $k=5$, and randomly select the neighbors from top-$K$ nearest neighbors to control the range of tokens' neighborhood. We show the Task-IL accuracies with different $K$ below:
| Wire-Neigh | DB | Yahoo |News Series
|-|-|-|-|
|  $K=5$  |   99.86  |   91.16  |76.90|
|  $K=20$  | 99.86|   90.98  |77.10|
|  $K=50$  | 99.86|  91.16 |77.20|
|  $K=100$  | 99.87 | 91.13|76.58|

For relatively simple sequences DB and Yahoo, Wire-Neigh under different $K$ has stable performance. However, for hard sequence News Series, when $K$ increases the model has more neighbor information (more capacity) to solve the task, which first improves its CL performance. However, when $K$ is too large ($K=100$), the neighbor information may become noisy, which makes the CL performance drop.
___

**4. Difference between Wire-Fixed, Wire-Neigh and C-LoRA.**

We illustrate the difference between alignment models based on Eq. 6, where aligned data representations ${\bf{h}}^{(l)}$ are expected to have the form $({\bf{h}}^{(l)})^T = ({\bf{b}}^{(l)})^T{\bf{A}}^{(l)}{\bf{G}}^{(l-1)}{\bf W}\_v^{(l)}$.
* **Wire-Fixed**: keep the pre-trained representations ${\bf{G}}^{(l-1)}{\bf W}\_v^{(l)}$ fixed, only learn the attention $ ({\bf{b}}^{(l)})^T{\bf{A}}^{(l)}$ by the self-attention mechanism with new key matrices, denoted as $SA(\Delta {\bf{W}}^{(l)}_k)$. This is only applied to [CLS] tokens. Then $({\bf{h}}^{(l)})^T = SA(\Delta {\bf{W}}^{(l)}_k){\bf{G}}^{(l-1)}{\bf W}\_v^{(l)}$. Wire-Fixed is strongly guided by pre-trained token representations but may have limited capacity in solving hard tasks.
* **Wire-Neigh**: apply the same wiring strategy as Wire-Fixed but add neighbor representations ${\bf{G}}\_{neigh}^{(l-1)}$ for better capacity. Then $({\bf{h}}^{(l)})^T = SA(\Delta {\bf{W}}^{(l)}_k)[{\bf{G}}^{(l-1)}; {\bf{G}}\_{neigh}^{(l-1)}]{\bf W}\_v^{(l)}$. The advantage of Wire-Neigh is it keeps the guidance of pre-trained token representations but increases the model capacity by expanding the neighborhood.
* **C-LoRA**: adapt both query and value matrices with $\Delta {\bf{W}}^{(l)}_q, \Delta {\bf{W}}^{(l)}_v$ but with a small scaling factor $s = 0.1$. This adaptation is applied to all tokens. Then $({\bf{h}}^{(l)})^T = SA({\bf{W}}^{(l)}_q + s\Delta {\bf{W}}^{(l)}_q){\bf{H}}^{(l-1)}({\bf W}\_v^{(l)}+s\Delta{\bf W}^{(l)}\_v)$, where ${\bf{H}}^{(l-1)}$ is the adapted token representation of the previous layer. Compared to wiring models, C-LoRA modifies both the attention and the value matrices applied to all tokens. It can still enjoy the guidance of pre-trained representations with the small scaling factor $s$, but the guidance may be weaker than wiring models. Meanwhile, it has more task-learning capacity. C-LoRA does not use neighbor information.
___

**5. The application of the methods to computer vision (CV) problems.**

* We believe our main contributions on alignment models and utilizing PF to reduce interference are general to CV tasks. First, our analysis of interference in Section 3 is general to both NLP and CV. Second, there are effective pre-trained models in CV as well, which can be used for alignment purposes.
* We think the keys to applying our model to CV tasks are: (1) how effective the pre-trained model is in providing self-supervised token representations for alignment; (2) how to properly set scaling ratios or select neighborhoods to balance the alignment effect and the task capacity on CV tasks.

We will study our models' application to CV in our future works.
___

**6. Typos and definitions in the writing.**

Thanks for pointing out the typos and the presentation suggestions. We will correct and modify the paper accordingly.

---

### Comment · Area_Chair_hjxC · 2024-08-11
**Discussion**

Dear Reviewers,

This paper has received diverging reviews ranging from 4 to 7. The authors have submitted a rebuttal and the anonymous link to their code https://drive.google.com/file/d/1PNjqBgr-ZUBmarNK-yPZ7GWUdFjNf7t0/view?usp=sharing. Most reviewers have raised their scores after reviewing the authors' rebuttal except one.

@Reviewer R2vv: Please read the authors' rebuttal/code and other reviewers' comments. Have the authors' responses addressed your concerns? If so, please reconsider the borderline decision.

AC

---

### Decision · Program_Chairs · 2024-09-25

**Decision:**

Accept (poster)

**Comment:**

This paper presents a global alignment method to alleviate catastrophic forgetting in continual learning. It first identifies the correlation between data representations is the main cause of gradient interference from different tasks. Then proposes a global alignment method to learn data representation as a composition of pre-trained token representations shared by all tasks, and explore different ways to learn such composition.  Extensive experiments demonstrate the effectiveness of the proposed method for both class-incremental and task-incremental learning.

The proposed approach has interesting theoretical justifications, and the code is provided for reproducing the experiments in the paper.

Reviewers raised some questions on why replacing the key matrix but not the query and value matrices in wiring models, and the authors explained the intuition behind the approach. Some reviewers also have concern on the extra computation induced by the alignment strategies and the PF. The authors' rebuttal addresses the above concern.

Most reviewers have raised their scores after reviewing the authors' rebuttal.
The reviewers find the technical novelty and contributions are significant enough for acceptance. The authors' rebuttal helps address some issues. The area chair agrees with the reviewers and recommend it be accepted.